# InvertiTune: Data Synthesis for Cost-Effective Single-Shot Text-to-Knowledge Graph Generation

**Faezeh Faez**                                                        *faezeh.faez@huawei.com*
*Huawei Noah's Ark Lab*

**Marzieh S. Tahaei**[*]                                       *marzieh.tahaei@mail.mcgill.ca*
*Autodesk*

**Yaochen Hu**                                                      *yaochen.hu@huawei.com*
*Huawei Noah's Ark Lab*

**Ali Pourranjbar**                                            *ali.pourranjbar@h-partners.com*
*Ascend Team, Huawei Technologies*

**Mahdi Biparva**                                                 *mahdi.biparva@huawei.com*
*Huawei Noah's Ark Lab*

**Mark Coates**                                                      *mark.coates@mcgill.ca*
*McGill University*

**Yingxue Zhang**                                              *yingxue.zhang@huawei.com*
*Huawei Noah's Ark Lab*

## Abstract

Large Language Models (LLMs) have revolutionized the ability to understand and generate text, enabling significant progress in knowledge graph construction from text (Text2KG). Many Text2KG methods, however, rely on iterative LLM prompting, making them computationally expensive and prone to overlooking complex relations distributed throughout the text. To address these limitations, we propose InvertiTune, a framework that combines a controlled data generation pipeline with supervised fine-tuning (SFT). Within this framework, the data-generation pipeline systematically extracts subgraphs from large knowledge bases, applies noise filtering, and leverages LLMs to generate corresponding natural text descriptions, a task more aligned with LLM capabilities than direct KG generation from text. This pipeline enables generating datasets composed of longer texts paired with larger KGs that better reflect real-world scenarios compared to existing benchmarks, thus supporting effective SFT of lightweight models for single-shot KG construction. Experimental results on CE12k, a dataset generated using our pipeline, show that InvertiTune outperforms larger non-fine-tuned LLMs as well as state-of-the-art Text2KG approaches, while demonstrating stronger cross-dataset generalization on CrossEval-1200, a test set created from three established benchmark datasets and CE12k. These findings highlight the importance of realistic training data for advancing efficient and high-performing Text2KG systems.

## 1 Introduction

Knowledge graphs (KGs) capture relationships between entities, making them valuable for applications such as question answering (Yih et al., 2015; Yani & Krisnadhi, 2021), clinical decision support (Cui et al.,

---

[*]Work done while at Huawei.

2025), recommendation systems (Zhang et al., 2016; Rong et al., 2024), logical reasoning (Chen et al., 2022), and fraud detection (Cai & Xie, 2024). Automatic KG construction from unstructured text (Text2KG) is desirable because manually curated graphs are costly to build and become outdated. Recent progress has been driven by iterative prompting of LLMs for KG construction, sometimes combined with additional steps such as verification, normalization, and refinement (Han et al., 2024; Chen et al., 2024; Bai et al., 2025; Huang et al., 2024). However, the reliance on extensive prompting can increase the computational cost of these approaches (Xu et al., 2023), and early mistakes may propagate through later stages (You et al., 2025), degrading the graph quality. One way to address these limitations is to train a model through supervised fine-tuning (SFT) that directly learns to map unstructured text to a structured knowledge graph representation in a single pass, thereby reducing the computational cost of iterative prompting and mitigating the risk of error propagation across stages.

One key challenge in advancing this direction is the lack of suitable datasets. Most well-known existing datasets typically consist of samples with relatively small KGs (Thanapalasingam et al., 2023; Mousavi et al., 2023), where the average number of triples per graph does not exceed five (see Table 1 for detailed statistics), and the corresponding texts are therefore short. Such simplified settings fail to capture the complexity of real-world scenarios, where the goal of KG construction is to represent intricate relationships across longer contexts, enabling deeper text understanding and more effective retrieval. Although some attempts (Choubey et al., 2024) have been made to build more realistic datasets, these KGs are synthesized through extensive multi-step prompting of LLMs to process the input text. Given the complexity of the task for current models, the resulting datasets may contain errors or incomplete information, which in turn raises concerns about their effectiveness in training improved Text2KG systems.

Motivated by these challenges, there is an evident need for appropriately sized datasets that can enable supervised fine-tuning of smaller models, potentially offering both efficiency and performance improvements. To this end, we introduce **InvertiTune**, a novel framework that addresses this gap through a controlled, goal-oriented data generation pipeline coupled with a supervised fine-tuning setup. Instead of extracting KGs from text, we invert the process: we first systematically extract subgraphs from a knowledge base (e.g., Wikidata), while incorporating multiple in-situ noise reduction steps into the extraction process, and then use a large language model to generate corresponding textual descriptions. This task is significantly more tractable for current LLMs and therefore less susceptible to inaccuracies. Once trained, our model generates knowledge graphs from texts in a single pass, eliminating costly iterative prompting and verification steps. Our key contributions are:

- We present a data generation pipeline that extracts semantically coherent subgraphs from a knowledge base and pairs them with LLM-generated textual descriptions to form reliable (text, KG) training instances.

- Using the data generation pipeline, we construct a 12k-sample dataset, **CE12k**, and demonstrate that it enables effective supervised fine-tuning of a lightweight LLM (Qwen 2.5-1.5B Instruct), which outperforms significantly larger counterparts as well as existing Text2KG baselines, offering both efficiency and performance improvements. The dataset is publicly available at `https://huggingface.co/datasets/FaezehFaez/CE12k`.

- We conduct cross-dataset generalization experiments on **CrossEval-1200**, a test set created from three established benchmark datasets as well as the **CE12k** dataset. We show that the model fine-tuned on our generated dataset achieves higher generalization than models trained on existing datasets, performing better on unseen data drawn from different distributions. The **CrossEval-1200** test set is publicly available at `https://huggingface.co/datasets/FaezehFaez/CrossEval-1200`.

- Using our data generation pipeline, we construct **CE-MultiCat-1200**, a challenging out-of-distribution test set in which initial entities are sampled from Wikidata categories disjoint from those used in **CE12k**, and show that the model trained on **CE12k** maintains strong performance under this distribution shift. The dataset is publicly available at `https://huggingface.co/datasets/FaezehFaez/CE-MultiCat-1200`.

- We perform several analyses on dataset scale and show that comparable performance can be achieved with fewer than 12k samples. This is considerably smaller than the number of samples in popular datasets, indicating that the quality of samples is more important than their quantity.

## 2  Related Work

We organize related work into two main categories: Text2KG methods, where the primary objective is to construct a structured knowledge graph directly from an input text, and Methods Generating KGs as Intermediate Representations, where graphs are also generated from text but serve mainly as intermediate representations to support broader applications such as multi-hop question answering or summarization.

### 2.1  Text2KG Methods

We further divide this category into approaches that operate without LLMs and those that leverage them.

**Non-LLM approaches.**  OpenIE6 (Kolluru et al., 2020) models relation extraction as iterative grid labeling over BERT representations, with training-time constraints to enhance recall and a coordination analyzer to address complex coordination structures. DeepEx (Wang et al., 2021) formulates triple extraction as a search problem. It first identifies token sequences relevant to a subject–object pair using attention signals from a pretrained language model, approximating the intractable exhaustive search via beam search. A ranking model trained on a task-agnostic relational corpus then makes the final decision. DetIE (Vasilkovsky et al., 2022) formulates relation extraction as a direct set prediction task inspired by object detection. It predicts a fixed number of possible triples in parallel and optimizes an order-agnostic bipartite matching loss to ensure unique predictions. Overall, while all these approaches are computationally efficient, they are confined to relations that are explicitly stated in text and often struggle to capture implicit semantics.

**LLM-based approaches.**  The rise of LLMs has led to prompt-driven frameworks for the Text2KG task. PiVe (Han et al., 2024) generates a knowledge graph from text by prompting an LLM and then employs a verifier, trained on a small pretrained LM, to iteratively refine it. The verifier inspects the generated graph, modifies the prompt when errors are detected, and re-queries the LLM until convergence. This feedback loop reduces noise but incurs the computational cost of repeated prompting and risks error propagation. EDC (Zhang & Soh, 2024) introduces a prompt-based framework that extracts triples, defines relation semantics through natural language descriptions, and refines the extracted relations to a canonical form using these definitions to eliminate redundancies and ambiguities. This multi-step process enhances consistency, but introduces prompting overhead. iText2KG (Lairgi et al., 2024) proposes a zero-shot pipeline that leverages LLM prompting for entity extraction, relation extraction, and integration, progressively updating the graph while avoiding duplication and inconsistency. The framework can be adapted to different domains through predefined schemas, but its reliance on repeated LLM calls introduces efficiency challenges. On the other hand, KGGEN (Mo et al., 2025) adopts a sequential series of clustering operations guided by LLM prompting to cluster semantically related entities, thereby mitigating sparsity in the extracted KGs. Within this line of work, GraphJudge (Huang et al., 2024) prompts a closed-source LLM to generate draft triples, from which positive and negative samples are constructed to perform supervised fine-tuning on an open-source LLM verifier. The verifier filters erroneous triples to improve KG quality. A limitation, however, is that the supervision itself comes from imperfect draft KGs, raising concerns of circularity and error reinforcement. AutoRE (Xue et al., 2024) adopts a supervised fine-tuning strategy as an alternative to the computationally expensive prompting-based frameworks in Text2KG research. It introduces the Relation–Head–Fact (RHF) paradigm, which decomposes document-level relation extraction into three subtasks: relation identification, head entity prediction, and factual triple generation. The authors design three tailored instructions, one for each step, and fine-tune an LLM on an instruction-finetuning dataset they crafted. This supervised approach enables efficient knowledge graph construction while alleviating the need for repeated LLM prompting.

## 2.2 Methods Generating KGs as Intermediate Representations

A second line of work views KG construction not as the ultimate goal but as a supporting step to enhance other applications. GraphRAG (Edge et al., 2024) builds a knowledge graph via LLM prompting for entity and relation extraction, applies community detection, and summarizes each community. Queries are then answered based on the indexed community summaries for query-focused summarization. LightRAG (Guo et al., 2024) uses LLM prompting to extract entities and relations from text and represents them as key–value pairs, where the key is a concise identifier and the value is a descriptive summary, enabling efficient retrieval. It employs a dual-level retrieval mechanism to handle both factual and abstract queries. GraphReader (Li et al., 2024) organizes long texts into a graph where each node corresponds to a normalized key element paired with its atomic facts, following the normalization approach of Lu et al. (2023). Nodes are connected when a key element of one appears in the atomic facts of another. Upon receiving a question, GraphReader employs an LLM-based agent that explores the graph to gather the necessary information to answer.

Despite progress, existing approaches remain constrained. Many methods either perform suboptimally or depend on repeated LLM prompting, which is computationally expensive and susceptible to error propagation. Furthermore, a persistent bottleneck in this area is the lack of reasonably sized text–KG datasets. This limitation hampers the adoption of more efficient and effective paradigms such as supervised fine-tuning, which could enable more accurate and cost-effective automatic KG construction from text.

# 3 Basic Definitions

## 3.1 Text2KG Task

A knowledge graph $\mathcal{G}$ is represented as a set of triples $\mathcal{G} = \{[s_i, p_i, o_i]\}_{i=1}^{N}$, where each triple $[s_i, p_i, o_i]$ consists of a subject $s_i$, predicate (relation) $p_i$, and object $o_i$. In the Text-to-Knowledge Graph (Text2KG) task, the goal is to construct a knowledge graph $\mathcal{G}$ from an input text $\mathcal{T}$ such that every triple $[s_i, p_i, o_i]$ in $\mathcal{G}$ is supported by the text. This means that the relation $p_i$ between $s_i$ and $o_i$ is either explicitly stated in $\mathcal{T}$ or can be implicitly inferred from it.

## 3.2 Data Generation for the Text2KG Task

The goal of data generation for the Text2KG task is to construct a dataset in which each sample $(\mathcal{T}, \mathcal{G})$ consists of a text $\mathcal{T}$ and its corresponding knowledge graph $\mathcal{G}$. Such a dataset enables the fine-tuning of large language models for this task. We assume access to a large auxiliary knowledge graph $\mathcal{G}_{\mathrm{aux}} = (\mathcal{E}, \mathcal{R}, \mathcal{F})$ that may contain noise or low-value information, where $\mathcal{E}$ represents the set of entities, $\mathcal{R}$ the set of relations, and $\mathcal{F} \subseteq \mathcal{E} \times \mathcal{R} \times \mathcal{E}$ the set of factual triples. Each entity in $\mathcal{E}$ may belong to a category $c$, forming a subset $\mathcal{E}_c \subseteq \mathcal{E}$ (e.g., $c$ may represent categories such as *Human*, *Disease*, *City*, or *Company*, among others). The generation of each data sample begins by extracting a denoised subgraph $\mathcal{G} = (\mathcal{E}', \mathcal{R}', \mathcal{F}')$ from $\mathcal{G}_{\mathrm{aux}}$, where $\mathcal{E}' \subseteq \mathcal{E}$, $\mathcal{R}' \subseteq \mathcal{R}$, and $\mathcal{F}' \subseteq \mathcal{F}$, followed by generating the corresponding textual description $\mathcal{T}$ for $\mathcal{G}$.

# 4 InvertiTune

We first present our pipeline for generating paired (text, KG) data, followed by a description of the supervised fine-tuning process. Figure 1 illustrates an overview of our InvertiTune framework with an example.

## 4.1 Data Generation Pipeline

The data generation pipeline begins with the controlled extraction of subgraphs from a large-scale knowledge base. Specifically, we introduce a recursive goal-oriented graph traversal strategy that ensures the semantic coherence of the extracted subgraphs while reducing noise through inline filtering mechanisms. These mechanisms enhance the quality of the extracted subgraphs at a negligible computational cost. Once subgraphs are obtained, the pipeline proceeds to generate corresponding textual descriptions for each subgraph.

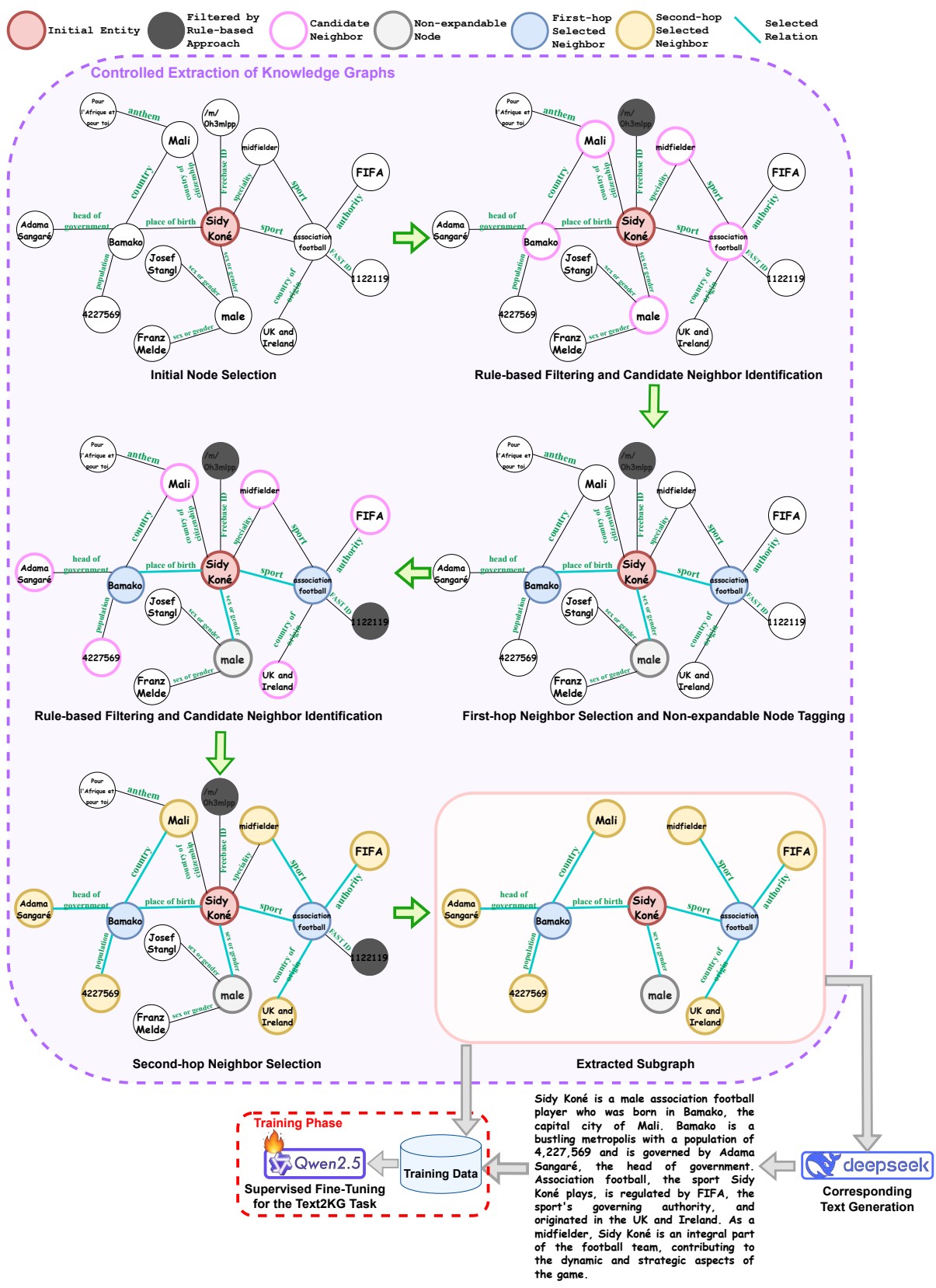

Figure 1: An overview of our InvertiTune framework, including the data generation pipeline and the training phase (SFT for the Text2KG task).

### 4.1.1 Controlled Knowledge Graph Extraction

Current knowledge bases are extremely large, comprising tens of millions of entities and relationships (Vrandečić et al., 2023). However, many triples are minimally informative or constitute noise. This makes extracting high-quality subgraphs challenging, as random graph traversal often yields subgraphs populated with low-value triples that lack semantic coherence. To address this issue, we introduce a controlled subgraph extraction process that generates informative and semantically coherent subgraphs through filtering constraints that suppress noise and control redundancy. Concretely, we begin extracting a subgraph $\mathcal{G} \subseteq \mathcal{G}_{\mathrm{aux}}$ by randomly sampling an initial entity $e_0 \in \mathcal{E}_c$ from a specified category $c$. We then identify the set of valid candidate triples originating from $e_0$ as

$$\mathcal{F}_{\mathrm{valid}}(e_0) = \big\{ [e_0, r, o] \in \mathcal{F} \mid \Phi_{\mathrm{noexpand}}(e_0) = 1, \ \Phi_{\mathrm{rule}}([e_0, r, o]) = 1, \ \Phi_{\mathrm{sp\text{-}uniq}}(e_0, r) = 1 \big\}.$$

Here, $\Phi_{\mathrm{noexpand}}$, $\Phi_{\mathrm{rule}}$, and $\Phi_{\mathrm{sp\text{-}uniq}}$ denote the operators corresponding to the entity expansion blacklist constraint, the rule-based validation filter, and the subject–predicate uniqueness constraint, respectively, which are applied sequentially, one after another. Specifically, $\Phi_{\mathrm{noexpand}}$ prevents exploring the neighbors of special entities, as including those neighbors in the subgraph would add only trivial information; $\Phi_{\mathrm{rule}}$ eliminates less informative or noisy triples through a set of simple, deterministic rules; and $\Phi_{\mathrm{sp\text{-}uniq}}$ enforces subject–predicate pair uniqueness, designed to make the extracted subgraphs compatible with prospective dataset generation for potential downstream applications such as multi-hop question answering. All three filtering mechanisms incur negligible computational cost and will be discussed in detail later.

After identifying the set of candidate triples $\mathcal{F}_{\mathrm{valid}}(e_0)$, we randomly select $m$ triples when $|\mathcal{F}_{\mathrm{valid}}(e_0)| > m$, and all are selected otherwise, yielding the retained set $\mathcal{F}^{(1)}(e_0) = \widehat{\mathcal{F}}_{\mathrm{valid}}(e_0) \subseteq \mathcal{F}_{\mathrm{valid}}(e_0)$. The corresponding first-hop neighbor entities are then

$$\mathcal{N}^{(1)}(e_0) = \{ o \mid [e_0, r, o] \in \widehat{\mathcal{F}}_{\mathrm{valid}}(e_0) \}.$$

The procedure is then recursively repeated at each subsequent hop $h$ ($2 \leq h \leq k$) for the entities identified in the previous hop. Specifically, for every entity $s \in \mathcal{N}^{(h-1)}(e_0)$, we determine its valid outgoing triples $\mathcal{F}_{\mathrm{valid}}(s)$ in a similar manner as for $e_0$, and then randomly select $m$ triples (or all triples if $m$ or fewer exist) to form the subset $\widehat{\mathcal{F}}_{\mathrm{valid}}(s) \subseteq \mathcal{F}_{\mathrm{valid}}(s)$. The selected triples and entities at hop $h$ are formalized as

$$\mathcal{F}^{(h)}(e_0) = \bigcup_{s \in \mathcal{N}^{(h-1)}(e_0)} \widehat{\mathcal{F}}_{\mathrm{valid}}(s), \quad \mathcal{N}^{(h)}(e_0) = \{ o \mid [s, r, o] \in \mathcal{F}^{(h)}(e_0) \}.$$

Continuing this recursive subgraph expansion, after adding the $k$-th-hop triples and entities (if available), we obtain the subgraph $\mathcal{G} = (\mathcal{E}', \mathcal{R}', \mathcal{F}')$, where

$$\mathcal{E}' = \{e_0\} \cup \bigcup_{h=1}^{k} \mathcal{N}^{(h)}(e_0), \quad \mathcal{F}' = \bigcup_{h=1}^{k} \mathcal{F}^{(h)}(e_0).$$

This systematic traversal ensures that each extracted subgraph contains coherent triples with direct or indirect relevance to the starting entity, facilitating the generation of semantically cohesive text in subsequent processing. The parameters $m$ and $k$ are configurable and can be adjusted to control the subgraph's size and depth. Furthermore, as alluded to earlier, to enhance triple quality while maintaining computational efficiency, filtering mechanisms are applied during extraction rather than as a post-processing step. These inline filters prevent non-qualified paths from being expanded, thereby excluding low-quality triples. Our pipeline employs three such mechanisms to ensure consistent knowledge extraction with improved quality.

### 4.1.1.1 Filtering Guided by the Entity Expansion Blacklist

Certain entities, while valuable to include in the extracted subgraph, should not be further expanded, as doing so adds only trivial information. Figure C.1 in Appendix C presents examples of trivial triples generated by expanding one of these entities. To address this, we curate a blacklist $\mathcal{B}$ of such entities using an LLM-assisted procedure, performed once as a preprocessing step before starting controlled knowledge graph extraction.

To construct $\mathcal{B}$, we first perform multiple graph traversals similar to the procedure described in Section 4.1.1. Each traversal begins from a randomly selected seed entity $e_0$, from which we extract its $k$-hop neighborhood. From the discovered neighbors, $\bigcup_{h=1}^{k} \mathcal{N}^{(h)}(e_0)$, we randomly sample a subset $\mathcal{S}_{e_0}$ and add these entities to a candidate set $\mathcal{C}$ containing entities potentially not worth expanding. This process is repeated with different initial seeds to ensure diversity within $\mathcal{C}$. Once $\mathcal{C}$ is constructed, we evaluate each of its elements to determine whether it should be included in $\mathcal{B}$. More precisely, for each candidate entity $e \in \mathcal{C}$, we retrieve all triples in which $e$ appears as the subject, forming the set $\mathcal{F}_e = \{[e, r, o]\}$. We then pass $\mathcal{F}_e$ through an LLM-based evaluator, denoted by $\phi_{\text{LLM}}$, which assesses whether all outgoing triples of $e$ are uninformative. If $\phi_{\text{LLM}}(\mathcal{F}_e) = 1$, the entity $e$ is added to the blacklist $\mathcal{B}$. Figure E.3 in Appendix E illustrates the prompt used for this evaluation, and the resulting curated entity expansion blacklist $\mathcal{B}$ is detailed in Appendix A.

Having established $\mathcal{B}$ in the initial setup phase, we incorporate it into the controlled knowledge graph extraction process to guide entity expansion decisions. Specifically, during each graph traversal, before expanding an entity $s$, we evaluate it using the operator $\Phi_{\text{noexpand}}$, where $\Phi_{\text{noexpand}}(s) = 1$ if $s \notin \mathcal{B}$ and 0 otherwise. This mechanism prevents unnecessary expansions that contribute little or no additional information, thereby maintaining the semantic quality of the extracted subgraph.

### 4.1.1.2  General Rule-Based Filtering

Public knowledge bases contain noisy or low-quality triples that provide limited semantic value. Many of these triples follow detectable patterns, such as containing URLs, identification numbers, or non-Latin characters. To address this issue, we design a set of filtering rules and apply them when selecting neighboring triples of a node $s$, such as $[s, r, o]$, to filter out such non-informative candidates. Let $\{r_1, \dots, r_K\}$ denote this set of rules, and define an operator $\Phi_{\text{rule}}$ that sequentially evaluates each rule on the triple $[s, r, o]$. Formally,

$$\Phi_{\text{rule}}([s, r, o]) = \begin{cases} 1, & \text{if } \forall j \in \{1, \dots, K\}, \ [s, r, o] \text{ satisfies } r_j, \\ 0, & \text{otherwise.} \end{cases}$$

Only triples satisfying $\Phi_{\text{rule}}([s, r, o]) = 1$ are retained for subsequent filtering. A description of the rules is provided in Appendix B. Figure C.2 in Appendix C presents examples of triples filtered by this mechanism.

### 4.1.1.3  Subject–Predicate Pair Uniqueness Enforcement

While the primary goal of our data generation pipeline is to produce $(\mathcal{T}, \mathcal{G})$ pairs for the Text2KG task, it is also designed to facilitate the creation of data for downstream applications such as multi-hop question answering over knowledge graphs. To support this prospective use case, we enforce subject–predicate pair uniqueness by applying the operator $\Phi_{\text{sp-uniq}}$ during subgraph extraction. Specifically, we exclude triples that share the same subject and predicate but differ in their object, as such cases would result in questions without a unique answer. Figure C.3 in Appendix C presents an example of such a violation along with a corresponding question that lacks a single definitive answer. Formally, the operator is defined as:

$$\Phi_{\text{sp-uniq}}(s, r) = \begin{cases} 0, & \text{if } \exists o, o' \in \mathcal{E}, \ o \neq o' \text{ such that } [s, r, o] \in \mathcal{F} \text{ and } [s, r, o'] \in \mathcal{F}, \\ & \quad \text{with } \Phi_{\text{rule}}([s, r, o]) = 1 \text{ and } \Phi_{\text{rule}}([s, r, o']) = 1, \\ 1, & \text{otherwise.} \end{cases}$$

Accordingly, a triple $[s, r, o]$ is retained only if $\Phi_{\text{sp-uniq}}(s, r) = 1$, meaning that among all triples passing the rule-based filtering step, there exists no other entity $o'$ such that $[s, r, o']$ also appears in the graph.

By following the steps described in Sections 4.1.1.1, 4.1.1.2, and 4.1.1.3, our extraction pipeline produces substantially higher-quality subgraphs, which serve as a strong foundation for the subsequent text generation phase. Figure C.4 in Appendix C presents examples of the resulting triples.

Table 1: Dataset Statistics Comparison

| Dataset | Split | # Samples | # Triples per KG | | | | # Tokens in Text | | | |
|---|---|---|---|---|---|---|---|---|---|---|
| | | | Min | Avg | Med | Max | Min | Avg | Med | Max |
| **KELM** | Train (97.09%) | 60,000 | 1 | 3.50 | 3.50 | 6 | 3 | 16.81 | 16.00 | 212 |
| | Test (2.91%) | 1,800 | 1 | 3.50 | 3.50 | 6 | 4 | 16.75 | 15.00 | 151 |
| **WebNLG+2020** | Train (95.22%) | 35,426 | 1 | 2.96 | 3.00 | 7 | 3 | 19.83 | 18.00 | 72 |
| | Test (4.78%) | 1,779 | 1 | 3.17 | 3.00 | 7 | 4 | 22.01 | 20.00 | 80 |
| **GenWiki-HIQ** | Train (99.09%) | 109,335 | 1 | 4.23 | 4.00 | 13 | 2 | 24.20 | 21.00 | 76 |
| | Test (0.91%) | 1,000 | 1 | 3.92 | 3.00 | 14 | 5 | 18.65 | 16.00 | 58 |
| **CE12k** | Train (90.00%) | 10,800 | 1 | 24.12 | 6.00 | 244 | 3 | 122.37 | 61.00 | 2909 |
| | Test (10.00%) | 1,200 | 1 | 25.01 | 7.50 | 207 | 4 | 126.46 | 67.00 | 536 |
| **CE12k-Relaxed** | Train (90.00%) | 10,800 | 1 | 98.73 | 11.00 | 915 | 3 | 289.87 | 88.00 | 6388 |
| | Test (10.00%) | 1,200 | 1 | 101.85 | 11.00 | 705 | 4 | 296.62 | 98.00 | 5596 |
| **CE-MultiCat-1200** | Train (0.00%) | 0 | - | - | - | - | - | - | - | - |
| | Test (100.00%) | 1,200 | 1 | 35.41 | 30.00 | 182 | 4 | 178.44 | 165.00 | 629 |
| **CrossEval-1200** | Train (0.00%) | 0 | - | - | - | - | - | - | - | - |
| | Test (100.00%) | 1,200 | 1 | 9.20 | 4.00 | 170 | 4 | 46.54 | 20.50 | 461 |
| **CrossEval-1500** | Train (0.00%) | 0 | - | - | - | - | - | - | - | - |
| | Test (100.00%) | 1,500 | 1 | 29.77 | 4.00 | 563 | 4 | 102.22 | 24.00 | 2090 |

### 4.1.2 Corresponding Text Generation

After subgraph extraction, we feed the triple sets from each subgraph to an LLM to generate corresponding textual descriptions. Figure E.2 in Appendix E illustrates the prompt used for text generation, and Figure C.5 in Appendix C presents an example description generated from the triple set in Figure C.4.

### 4.2 Supervised Fine-tuning of Lightweight LLMs

After completing the data generation process, where each sample consists of a (text, KG) pair, we use the resulting dataset to perform SFT on a lightweight LLM. Once fine-tuning is complete, the model can be used at inference time by providing an input text, and it generates the corresponding knowledge graph in a single pass by outputting the associated set of triples.

## 5 Experiments

This section presents our evaluation of InvertiTune using established metrics and state-of-the-art baselines. The primary objective of these experiments is to assess the effectiveness of the proposed data generation pipeline by evaluating whether the generated datasets can support supervised fine-tuning (SFT) for the Text2KG task, thereby providing an indirect measure of their usefulness.

### 5.1 Experimental Setup

**Datasets.** Using our data generation pipeline with controlled extraction, we created a dataset of 12,000 samples, which we refer to as CE12k. Specifically, we generated 6,000 samples with $m = 4$ neighboring nodes and $k = 6$ hops, 2,000 samples with $m = 6$ and $k = 1$, 2,000 samples with $m = 2$ and $k = 3$, and 2,000 samples with $m = 3$ and $k = 2$, resulting in a total of 12,000 samples. Importantly, our approach is a pipeline, enabling the generation of datasets with arbitrary sample counts and graph sizes by varying the values of $m$ and $k$. We also constructed an additional dataset, CE12k-Relaxed, and three test sets, CE-MultiCat-1200, CrossEval-1200, and CrossEval-1500, which are discussed in detail later in the paper. Furthermore, we conducted experiments on well-known existing Text2KG datasets, namely KELM, WebNLG+2020, and GenWiki-HIQ. Key statistics of these datasets are summarized in Table 1, and illustrative visualizations of some of their distributions are provided in Appendix D.

**Evaluation Metrics.** We use standard evaluation metrics for the Text2KG task. **G-BLEU (G-BL)** (Huang et al., 2024) evaluates knowledge graph predictions by representing each triple as a sequence and

computing BLEU-based similarity scores between predicted and reference triples. The evaluation matches triples from the two graphs to maximize the overall similarity and reports the F1 score. **G-ROUGE (G-RO)** (Huang et al., 2024) follows a procedure similar to G-BLEU, but employs ROUGE-based similarity. The resulting F1 score is reported. **BERTScore** (Zhang et al., 2019) is used in experiments where outputs are not necessarily graph-structured; in such cases, we compute it between the method output and the ground truth, treating them as text. **G-BERTScore (G-BS)** (Saha et al., 2021) extends BERTScore (Zhang et al., 2019) to graph structures by finding the optimal alignment between predicted and ground-truth edges. It treats each edge as a sentence, computes BERTScore between aligned pairs, and reports the F1 score.

**Baselines.** To evaluate the performance of our proposed approach against existing methods, we conduct extensive experiments with baselines from different categories. OpenIE6 (Kolluru et al., 2020) and DeepEx (Wang et al., 2021) are non-LLM approaches that rely on rule-based or BERT-based techniques. These methods are comparatively less expensive since they do not involve LLMs. PiVe (Han et al., 2024) is an approach specifically designed for the Text2KG task. It prompts an LLM to generate candidate KGs, then employs a verifier to identify problematic outputs and refine the prompt. This process is repeated iteratively to progressively improve KG quality. GraphRAG (Edge et al., 2024) and LightRAG (Guo et al., 2024) are also prompt-based methods; however, unlike PiVe (Han et al., 2024), they construct knowledge graphs from text as intermediate representations to facilitate downstream tasks. Such LLM-based prompting methods are generally more expensive, as they rely on multiple prompts to construct graphs. We also include ChatGPT as a baseline, where the text is directly provided to the model to generate the corresponding KG; the prompt used is shown in Figure E.1 in Appendix E. Beyond prompting-based methods, we also include AutoRE (Xue et al., 2024), a supervised fine-tuning approach designed for document-level relation extraction. It fine-tunes Mistral-7B using QLoRA on a instruction-finetuning dataset derived from Re-DocRED, following the Relation–Head–Fact (RHF) paradigm. Including AutoRE allows for a more comprehensive and fair comparison with our SFT-based approach, as both methods rely on fine-tuning rather than repeated LLM prompting. Finally, we compare the performance of our proposed **InvertiTune** approach to the Qwen2.5-1.5B Instruct and Qwen2.5-32B Instruct models to evaluate both efficiency and effectiveness.

**Implementation Details.** We use Wikidata as the knowledge base for subgraph extraction. The initial entities (e.g., $e_0$) are randomly sampled from the category $c = Human$ (i.e., $e_0 \in \mathcal{E}_{Human}$). The textual description $\mathcal{T}$ for each extracted subgraph $\mathcal{G}$ is generated using DeepSeek-V3. For supervised fine-tuning, we employ the Qwen2.5-1.5B Instruct model, chosen for its balance between efficiency and performance.

## 5.2 Experimental Results

To assess the effectiveness of our data generation pipeline for producing training data suitable for supervised fine-tuning in the Text2KG task, we conducted a series of experiments from four perspectives. First, in Section 5.2.1, we evaluate the performance of our InvertiTune model, comparing it to state-of-the-art Text2KG baselines and showing that it consistently outperforms them. Second, in Section 5.2.2, we analyze parameter efficiency and demonstrate that fine-tuning a lightweight model on suitable data is far more efficient and effective than relying on much larger models. Third, in Section 5.2.3, we examine cross-dataset generalization by fine-tuning the same model on our pipeline-generated dataset and on existing Text2KG datasets, showing that the former yields better generalization to unseen data from both the same and different distributions. Finally, in Section 5.2.4, we study the impact of dataset scale and show that even with fewer samples, the fine-tuned model can achieve competitive performance.

### 5.2.1 Comparison with Baselines

To evaluate the effectiveness of the dataset generated through our data generation pipeline for model fine-tuning in the Text2KG task, we compare the performance of the InvertiTune model (Qwen 2.5-1.5B Instruct fine-tuned on the training set of the CE12k dataset) against several baselines on the CE12k test set. The results are presented in Table 2. As shown, InvertiTune consistently outperforms the baselines across all metrics by a substantial margin, demonstrating that with suitable training data, even a lightweight model such as Qwen 2.5-1.5B Instruct can achieve strong performance. We also report 95% bootstrapped confidence intervals across 10,000 iterations to quantify the variability of each metric across test examples. To assess the statistical significance of InvertiTune's superior performance, we apply the Wilcoxon signed-rank test on

Table 2: Performance comparison of InvertiTune against a broad set of competing approaches on the CE12k test set. The best results are shown in **bold** and the second-best are underlined. The 95% bootstrapped confidence intervals are computed using 10,000 resampling iterations, and $p$-values are obtained from the Wilcoxon signed-rank test comparing InvertiTune vs. each baseline.

| Method | G-BLEU (G-BL) | | | G-ROUGE (G-RO) | | | G-BERTScore (G-BS) | | |
|---|---|---|---|---|---|---|---|---|---|
| | Score | 95% CI | $p$-value | Score | 95% CI | $p$-value | Score | 95% CI | $p$-value |
| OpenIE6 | 4.02 | (3.70, 4.36) | 8.0268e-198 | 6.51 | (5.94, 7.10) | 5.2556e-199 | 41.33 | (39.77, 42.90) | 1.5361e-194 |
| DeepEx | 6.32 | (6.08, 6.56) | 8.1335e-198 | 12.53 | (12.09, 12.99) | 1.1829e-197 | 52.97 | (51.91, 54.05) | 1.4788e-197 |
| PIVE | 39.04 | (38.08, 40.02) | 1.0465e-196 | 48.34 | (47.37, 49.28) | 2.6441e-196 | 75.06 | (74.15, 75.96) | 4.7978e-170 |
| AutoRE | 26.31 | (25.56, 27.09) | 8.3608e-197 | 30.83 | (30.06, 31.60) | 8.8932e-197 | 67.14 | (66.26, 68.04) | 9.3498e-182 |
| ChatGPT | 31.51 | (30.56, 32.49) | 1.4274e-197 | 40.22 | (39.10, 41.33) | 3.6104e-197 | 71.54 | (70.46, 72.59) | 3.9489e-180 |
| GraphRAG | 7.03 | (6.75, 7.31) | 8.1277e-198 | 9.03 | (8.72, 9.35) | 7.8674e-198 | 51.19 | (49.98, 52.36) | 6.2660e-197 |
| LightRAG | 4.82 | (4.63, 5.02) | 8.1268e-198 | 6.90 | (6.66, 7.15) | 7.7449e-198 | 51.81 | (50.57, 53.06) | 1.5049e-197 |
| InvertiTune (Ours) | **82.02** | (81.03, 83.01) | - | **82.67** | (81.69, 83.66) | - | **92.58** | (92.08, 93.06) | - |

Table 3: Performance comparison of InvertiTune against competing approaches on the CE-MultiCat-1200 test set. The best results are shown in **bold** and the second-best are underlined. Confidence intervals and $p$-values are computed following the same protocol as in Table 2.

| Method | Average Inference Time per Sample (s) | G-BLEU (G-BL) | | | G-ROUGE (G-RO) | | | G-BERTScore (G-BS) | | |
|---|---|---|---|---|---|---|---|---|---|---|
| | | Score | 95% CI | $p$-value | Score | 95% CI | $p$-value | Score | 95% CI | $p$-value |
| OpenIE6 | 52.71 | 10.74 | (10.43, 11.07) | 8.1951e-198 | 18.86 | (18.41, 19.32) | 8.5483e-198 | 73.42 | (72.76, 74.07) | 2.0502e-107 |
| DeepEx | 126.33 | 5.70 | (5.42, 5.99) | 8.1957e-198 | 10.73 | (10.30, 11.19) | 8.1910e-198 | 44.39 | (43.49, 45.29) | 4.2025e-197 |
| PIVE | **4.65** | 33.52 | (32.51, 34.57) | 4.7808e-183 | 41.22 | (40.17, 42.22) | 5.3608e-172 | 69.21 | (68.28, 70.11) | 9.2927e-122 |
| AutoRE | 19.56 | 26.16 | (25.48, 26.87) | 4.8962e-195 | 30.05 | (29.35, 30.77) | 6.8069e-194 | 69.40 | (68.55, 70.23) | 5.1007e-116 |
| ChatGPT | 7.18 | 24.95 | (24.34, 25.56) | 2.6480e-196 | 32.75 | (32.00, 33.51) | 1.1068e-193 | 64.89 | (63.90, 65.85) | 3.8868e-159 |
| GraphRAG | 42.11 | 6.01 | (5.81, 6.22) | 8.1531e-198 | 7.67 | (7.42, 7.93) | 8.0603e-198 | 44.19 | (43.16, 45.24) | 4.6775e-195 |
| LightRAG | 24.1 | 4.51 | (4.36, 4.67) | 8.1340e-198 | 6.29 | (6.09, 6.49) | 8.1156e-198 | 46.04 | (44.91, 47.15) | 5.3188e-194 |
| InvertiTune (Ours) | 9.05 | **62.20** | (61.12, 63.28) | - | **63.38** | (62.31, 64.42) | - | **84.00** | (83.36, 84.63) | - |

paired samples comparing InvertiTune with each baseline. The small $p$-values reported in Table 2 confirm that the performance gains are significant and not due to random variation. Moreover, to provide a sense of the outputs produced by different models, we present their results on a sample from the CE12k test set in Appendix F.

Beyond the CE12k test set, we further evaluate all methods on a test set whose distribution differs from the training data of InvertiTune. Specifically, we construct CE-MultiCat-1200, a test set in which the initial entity $e_0$ is sampled from four diverse Wikidata categories, namely *Mountain*, *University*, *Film*, and *City* (i.e., $e_0 \in \mathcal{E}_{Mountain} \cup \mathcal{E}_{University} \cup \mathcal{E}_{Film} \cup \mathcal{E}_{City}$), deliberately excluding the category $c = Human$ from which the initial entities in CE12k are sampled to ensure a distribution shift. Each category contributes 300 samples, yielding 1,200 test instances in total, where subgraphs are extracted with $m = 4$ neighboring nodes and $k = 6$ hops. The statistics of this test set are reported in Table 1. This setting provides a stricter evaluation under a more challenging out-of-distribution scenario. As reported in Table 3, InvertiTune consistently outperforms all baselines across all three metrics by a substantial margin, confirming that its superiority is not an artifact of distributional overlap between training and test data. The table also reports the average inference time per sample for each method as a reference point for latency. The confidence intervals and statistical significance reported in Table 3 follow the same protocol as in Table 2.

### 5.2.2 Parameter Efficiency Analysis

To evaluate the efficiency of our approach, we compare the performance of the InvertiTune model against the Qwen2.5-1.5B Instruct and Qwen2.5-32B Instruct models. As shown in Table 4, InvertiTune significantly outperforms both baselines. Notably, the much larger Qwen2.5-32B Instruct model provides only a marginal improvement over the 1.5B variant, yet still falls well short of InvertiTune. These results highlight the critical role of training data tailored for the Text2KG task: even a lightweight model like Qwen2.5-1.5B Instruct, when fine-tuned on such data, can outperform much larger models. We report results using BERTScore

Table 4: BERTScore evaluation on the CE12k test set showing that InvertiTune (1.5B) delivers the highest precision, recall, and F1 while using far fewer parameters than the 32B baseline, demonstrating strong parameter efficiency enabled by Text2KG training data generated by our pipeline.

| Model | Precision | Recall | F1 |
|---|---|---|---|
| InvertiTune (Ours) | **95.77** | **95.70** | **95.73** |
| Qwen2.5-1.5B Instruct | 81.98 | 82.92 | 82.43 |
| Qwen2.5-32B Instruct | 82.71 | 84.46 | 83.57 |

Table 5: Performance of Qwen2.5-1.5B Instruct model fine-tuned on different training sets, evaluated on the CrossEval-1200 test set. The best results are shown in **bold** and second-best are underlined. The 95% bootstrapped confidence intervals are computed using 10,000 iterations, and $p$-values are obtained from the Wilcoxon signed-rank test comparing the model fine-tuned on our CE12k training set against the models fine-tuned on alternative training sets.

| Training Dataset | G-BLEU (G-BL) | | | G-ROUGE (G-RO) | | | G-BERTScore (G-BS) | | |
|---|---|---|---|---|---|---|---|---|---|
| | Score | 95% CI | $p$-value | Score | 95% CI | $p$-value | Score | 95% CI | $p$-value |
| KELM | 48.21 | (46.40, 50.01) | 2.2880e-05 | 52.61 | (50.87, 54.34) | 9.8433e-08 | 79.21 | (77.86, 80.59) | 1.5741e-05 |
| WebNLG+2020 | 38.36 | (36.66, 40.15) | 4.0500e-25 | 43.55 | (41.84, 45.28) | 4.6036e-30 | 79.11 | (78.05, 80.15) | 6.8679e-21 |
| GenWiki-HIQ | 35.80 | (34.36, 37.30) | 1.6490e-32 | 45.31 | (43.89, 46.73) | 9.7687e-25 | 75.88 | (74.59, 77.17) | 4.1843e-23 |
| CE12k (Ours) | **52.65** | (51.01, 54.32) | - | **58.40** | (56.92, 59.96) | - | **85.19** | (84.32, 85.99) | - |

metrics, as the outputs of the baseline models do not consistently follow a structured KG (set-of-triples) format.

### 5.2.3 Cross-Dataset Generalization

A strength of fine-tuned models lies in their ability to generalize to unseen data. While they typically perform well on data drawn from distributions similar to the training set, their performance often degrades when faced with different distributions. To assess whether our generated dataset for supervised fine-tuning enables models to achieve stronger cross-dataset generalization than existing datasets, we conduct a series of evaluations.

To this purpose, we first constructed a test set of 1,200 samples, which we refer to as **CrossEval-1200**. This set was created by randomly selecting an equal number of samples from the test sets of KELM, WebNLG+2020, GenWiki-HIQ, and CE12k. The statistics of **CrossEval-1200** are presented in Table 1, and some distributional visualizations are provided in Figure D.7 in Appendix D.

We then fine-tuned the Qwen2.5-1.5B Instruct model separately on the training set of each dataset and evaluated its performance on **CrossEval-1200**. The results in Table 5 demonstrate that the model fine-tuned on our dataset achieves the highest performance, indicating that our dataset not only enables strong in-domain performance but enhances cross-domain generalization. To support this conclusion, we report 95% bootstrapped confidence intervals and Wilcoxon signed-rank tests to ensure the reliability of improvements.

To provide a stricter assessment of generalization, we repeat the same evaluation on CE-MultiCat-1200 (see Section 5.2.1 for details), a test set whose distribution differs from all training sets used in this comparison. Unlike CrossEval-1200, where part of the test samples share a similar distribution with the training data of each model, CE-MultiCat-1200 presents a more challenging out-of-distribution scenario for all models. As reported in Table 6, the model fine-tuned on CE12k continues to outperform all counterparts by a substantial margin, with the performance gap being even more evident than in the CrossEval-1200 evaluation, further confirming that CE12k enables stronger and more transferable generalization.

Table 6: Performance of Qwen2.5-1.5B Instruct model fine-tuned on different training sets, evaluated on the CE-MultiCat-1200 test set. Statistical reporting follows Table 5.

| Training Dataset | G-BLEU (G-BL) | | | G-ROUGE (G-RO) | | | G-BERTScore (G-BS) | | |
|---|---|---|---|---|---|---|---|---|---|
| | Score | 95% CI | *p*-value | Score | 95% CI | *p*-value | Score | 95% CI | *p*-value |
| KELM | 24.50 | (23.27, 25.77) | 4.7191e-185 | 26.89 | (25.60, 28.19) | 1.6625e-183 | 40.17 | (38.72, 41.64) | 6.2286e-184 |
| WebNLG+2020 | 16.19 | (15.46, 16.96) | 1.1498e-195 | 20.39 | (19.55, 21.24) | 2.6882e-195 | 53.46 | (52.39, 54.54) | 2.4852e-187 |
| GenWiki-HIQ | 14.01 | (13.17, 14.90) | 1.0626e-194 | 17.53 | (16.63, 18.47) | 4.5957e-194 | 35.10 | (33.74, 36.48) | 1.8773e-192 |
| CE12k (Ours) | 62.20 | (61.12, 63.28) | - | 63.38 | (62.31, 64.42) | - | 84.00 | (83.36, 84.63) | - |

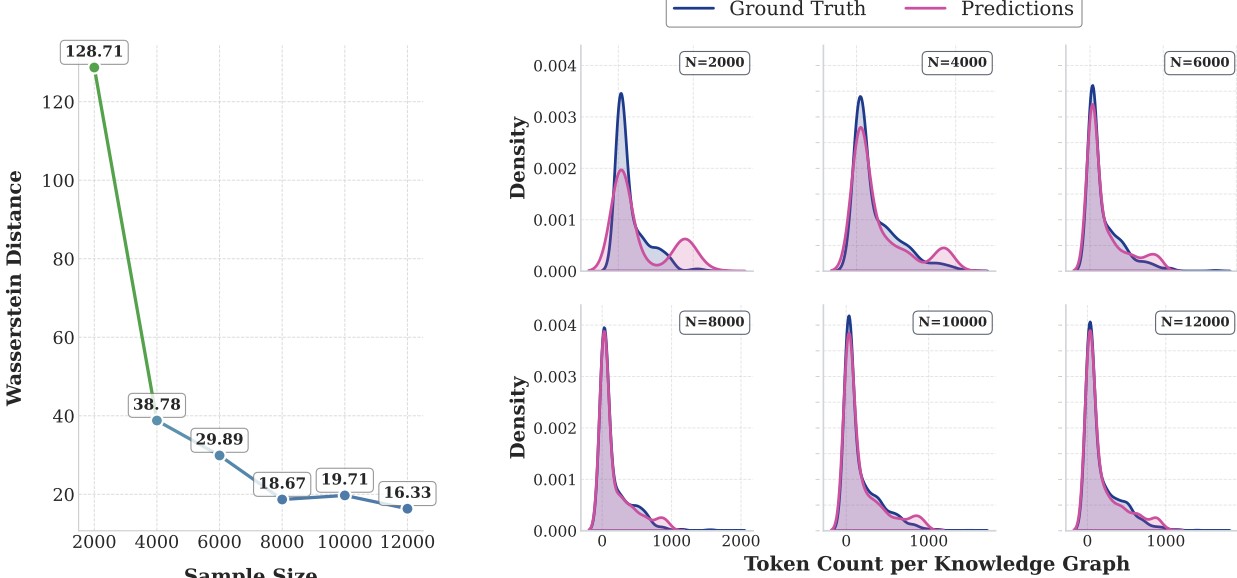

Figure 2: Convergence of token count distributions between predicted and ground-truth knowledge graphs across dataset sizes ($N = 2000$–$12000$). The left plot shows the Wasserstein distance, while the right panels present kernel density estimates, indicating closer alignment as $N$ increases.

### 5.2.4 Dataset Scale Analysis

We next examine the impact of dataset scale on model performance by progressively increasing the dataset size. Specifically, the number of samples was varied from 2,000 to 12,000 in increments of 2,000, with each dataset randomly drawn from the main corpus. For each dataset, the Qwen2.5–1.5B Instruct model was fine-tuned on 90% of the data and evaluated on the remaining 10%. All results reported in this section correspond to performance on the respective 10% test set of each dataset.

We begin our analysis by examining how the predicted knowledge graphs converge the ground truth with respect to the token count per knowledge graph and the triple count per knowledge graph. Figure 2 presents the convergence of token count distributions, while Figure 3 reports the analysis for triple count distributions. In both cases, larger dataset sizes typically lead to closer alignment between predicted and ground-truth distributions. The results further suggest a saturation trend, indicating that strong performance can be achieved even with moderately sized datasets, while additional scaling yields only limited improvements.

To further analyze the effect of dataset scaling and to examine model behavior from another perspective, we investigate how the performance gap between InvertiTune and the Qwen2.5–1.5B Instruct model evolves as a function of the number of samples. The results in Table 7 and Figure 4 are reported using the BERTScore metrics between each model's predictions and the ground truth. These results reveal a trend consistent with our earlier findings: as the number of samples increases, the performance gap typically widens. Nevertheless, a saturation point is again observed, indicating that smaller datasets with fewer than the maximum available

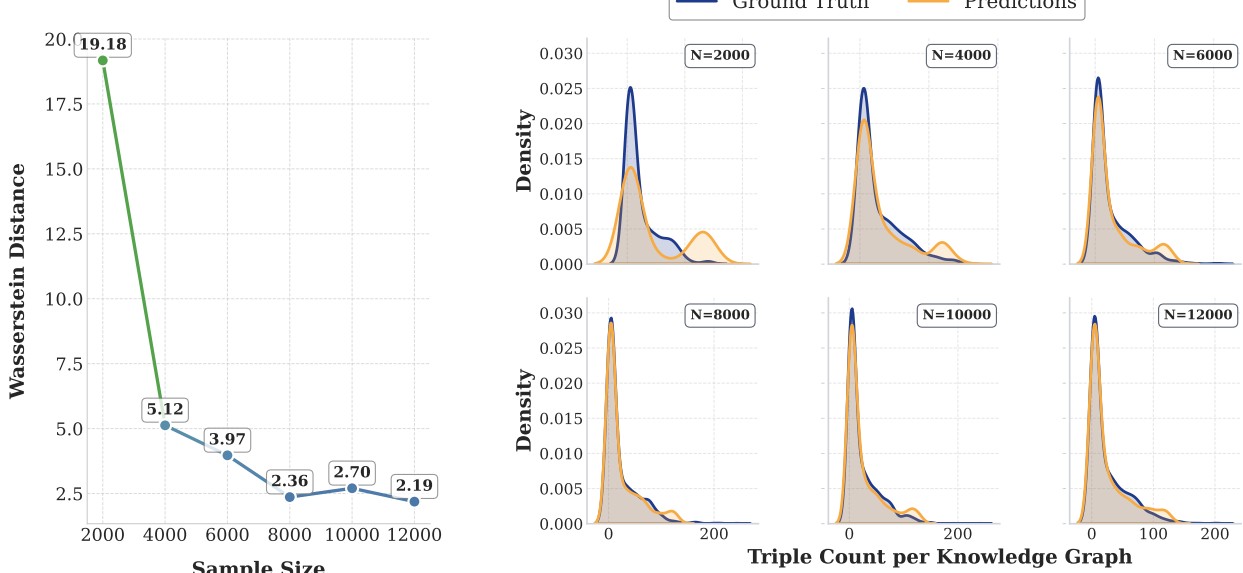

Figure 3: Convergence of triple count distributions between predicted and ground-truth knowledge graphs across dataset sizes ($N = 2000$–$12000$). The left plot shows the Wasserstein distance, while the right panels present kernel density estimates, with convergence becoming more evident as $N$ increases.

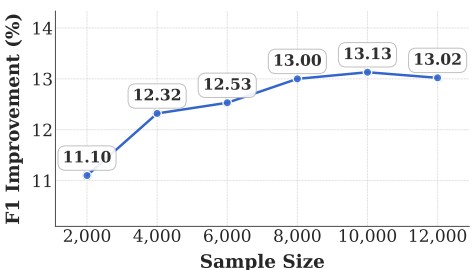

Figure 4: BERTScore F1 improvement of the InvertiTune model over the Qwen2.5-1.5B Instruct model as a function of sample size. Values are derived from the statistics reported in Table 7.

| Sample Size | 2000 | 4000 | 6000 | 8000 | 10000 | 12000 |
|---|---|---|---|---|---|---|
| **Precision** | | | | | | |
| InvertiTune | 93.95 | 94.90 | 95.33 | 95.82 | 95.84 | 95.73 |
| Qwen2.5-1.5B Instruct | 82.23 | 82.21 | 82.45 | 82.40 | 82.33 | 82.33 |
| Δ | +11.72 | +12.69 | +12.88 | +13.42 | +13.51 | +13.40 |
| **Recall** | | | | | | |
| InvertiTune | 93.63 | 94.70 | 95.22 | 95.77 | 95.76 | 95.63 |
| Qwen2.5-1.5B Instruct | 83.18 | 82.78 | 83.07 | 83.23 | 83.06 | 83.03 |
| Δ | +10.45 | +11.92 | +12.15 | +12.54 | +12.70 | +12.60 |
| **F1** | | | | | | |
| InvertiTune | 93.78 | 94.80 | 95.27 | 95.80 | 95.80 | 95.68 |
| Qwen2.5-1.5B Instruct | 82.68 | 82.48 | 82.74 | 82.80 | 82.67 | 82.66 |
| Δ | +11.10 | +12.32 | +12.53 | +13.00 | +13.13 | +13.02 |

Table 7: BERTScore performance comparison (Precision, Recall, and F1) between InvertiTune and Qwen2.5-1.5B Instruct across different sample sizes.

samples can still achieve comparable performance. We report these results using the BERTScore metrics, as the outputs of Qwen2.5–1.5B Instruct do not necessarily conform to a structured knowledge graph format.

We finally examined the trend of InvertiTune's performance with increasing dataset size using KG-based evaluation metrics. The results, shown in Figure 5, confirm the earlier conclusion regarding the existence of an optimal dataset size and reveal a similar saturation pattern.

The results demonstrate that fine-tuning on the full 12K-sample dataset is not required to achieve optimal or near-optimal performance. Comparable outcomes can be obtained with as few as 8K to 10K samples, which is substantially fewer than datasets previously used for the text-to-KG task, as summarized in Table 1.

### 5.2.5 Ablation on the Subject–Predicate Pair Uniqueness Constraint

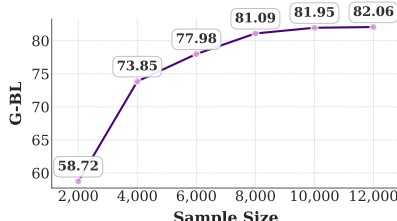 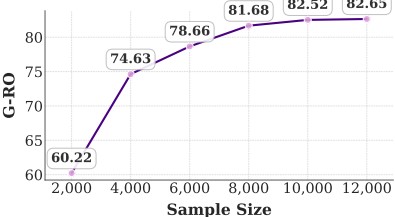 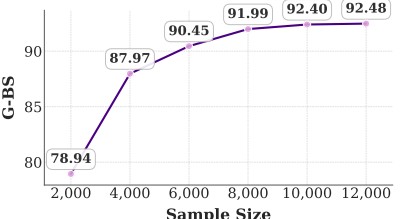

Figure 5: Performance of InvertiTune under SFT across varying dataset sizes, evaluated using KG-based metrics.

Table 8: Comparison of models fine-tuned on CE12k and CE12k-Relaxed across multiple test sets. Statistical reporting follows Table 5.

| Test Dataset | Training Dataset | G-BLEU (G-BL) | | | G-ROUGE (G-RO) | | | G-BERTScore (G-BS) | | |
|---|---|---|---|---|---|---|---|---|---|---|
| | | Score | 95% CI | $p$-value | Score | 95% CI | $p$-value | Score | 95% CI | $p$-value |
| KELM | CE12k-Relaxed | 55.00 | (53.82, 56.12) | 2.0434e-03 | 62.55 | (61.49, 63.60) | 6.9506e-02 | 81.17 | (80.44, 81.86) | 9.6804e-01 |
| | CE12k | 53.87 | (52.74, 55.04) | - | 62.21 | (61.17, 63.26) | - | 81.71 | (81.05, 82.37) | - |
| WebNLG+2020 | CE12k-Relaxed | 32.88 | (31.88, 33.90) | 3.3685e-43 | 41.24 | (40.15, 42.29) | 1.0153e-46 | 79.66 | (78.72, 80.54) | 3.6314e-40 |
| | CE12k | 38.55 | (37.51, 39.61) | - | 47.59 | (46.50, 48.67) | - | 86.16 | (85.56, 86.74) | - |
| GenWiki-HIQ | CE12k-Relaxed | 29.30 | (28.07, 30.52) | 1.2281e-01 | 33.73 | (32.49, 34.97) | 2.8034e-02 | 78.74 | (77.67, 79.78) | 9.2611e-04 |
| | CE12k | 29.83 | (28.57, 31.08) | - | 34.87 | (33.58, 36.17) | - | 77.49 | (76.41, 78.56) | - |
| CrossEval-1500 | CE12k-Relaxed | 49.11 | (47.72, 50.53) | 1.8172e-10 | 53.30 | (51.92, 54.64) | 8.3865e-12 | 80.16 | (79.26, 81.10) | 1.6300e-09 |
| | CE12k | 52.66 | (51.22, 54.10) | - | 57.38 | (55.99, 58.78) | - | 83.57 | (82.78, 84.37) | - |

As described earlier, our subgraph filtering stage enforces subject–predicate pair uniqueness to support potential downstream applications. However, since this constraint may affect the distribution of the extracted subgraphs, it is important to ensure that it does not degrade the effectiveness of the resulting training data.

To examine this question, we construct a variant of our dataset, denoted **CE12k-Relaxed**, by disabling the subject–predicate uniqueness constraint while keeping the rest of the pipeline unchanged. CE12k-Relaxed is generated using the same procedure as CE12k, as described in Section 5.1, with its statistics reported in Table 1. We then fine-tune Qwen2.5-1.5B Instruct separately on CE12k and CE12k-Relaxed under identical training conditions, and evaluate both models on four test sets. Three of these, KELM, GenWiki-HIQ, and WebNLG+2020, are benchmark datasets that are distributionally distinct from both CE12k and CE12k-Relaxed. We also construct **CrossEval-1500**, an extended version of CrossEval-1200 obtained by augmenting it with 300 randomly selected samples from the CE12k-Relaxed test split, enabling a fair comparison. The CE12k-Relaxed dataset is publicly available at https://huggingface.co/datasets/FaezehFaez/CE12k-Relaxed, while CrossEval-1500 is available at https://huggingface.co/datasets/FaezehFaez/CrossEval-1500.

Table 8 summarizes the results of this comparison, showing that the model fine-tuned on CE12k performs comparably and in some cases slightly better than the model fine-tuned on CE12k-Relaxed. Specifically, on KELM and GenWiki-HIQ, the two models perform similarly. On WebNLG+2020 and CrossEval-1500, the model trained on CE12k achieves modest improvements. Taken together, these findings indicate that enforcing subject–predicate pair uniqueness does not adversely affect the effectiveness of the resulting training data for Text2KG.

## 6 Conclusion

We presented a framework that combines a data-generation pipeline with a supervised fine-tuning setup for the Text2KG task. By extracting noise-minimized, semantically coherent subgraphs from a large, inherently noisy knowledge base and generating corresponding texts using a large language model, our pipeline can produce effective training sets of varying sizes and statistical characteristics for supervised fine-tuning. Ex-

periments demonstrated that **InvertiTune**, obtained by fine-tuning the lightweight Qwen2.5-1.5B Instruct model on **CE12k**, a dataset generated by our pipeline, outperforms much larger non-fine-tuned models as well as existing baselines for automatic knowledge graph construction from text. These findings underscore the value of high-quality data generation combined with a simple training setup. Future work will explore alternative model architectures and the use of diverse LLMs for text generation.

## Limitations

Our work has limitations. The evaluation was restricted to the Qwen2.5-1.5B Instruct model, and larger or different architectures were not examined. Moreover, since our approach relies on large language models to generate textual descriptions from knowledge graphs, and we only used DeepSeek-V3 for this step, assessing the impact of alternative LLMs on output quality and potential improvements remains an open direction. Furthermore, as with any automated data generation approach, the resulting data may contain errors or omissions in the absence of human verification. In addition, since the dataset is constructed from a subset of Wikidata, it may reflect biases or coverage limitations of that source. These aspects should be taken into account when using the dataset, particularly in fully automated knowledge base construction settings.

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

# A   Entity Expansion Blacklist

The entity expansion blacklist $\mathcal{B}$, used within one of the three inline filtering mechanisms during controlled knowledge graph extraction, is presented in Table A.1. Additional entities can be incorporated into $\mathcal{B}$ by performing further graph traversals following the same procedure described in Section 4.1.1.1.

| Entity ID | Entity Name | Entity ID | Entity Name |
|---|---|---|---|
| Q6581097 | male | Q4164871 | position |
| Q5 | human | Q192581 | job activity |
| Q12308941 | male given name | Q268378 | work |
| Q51929218 | first-person singular | Q486972 | human settlement |
| Q51929403 | second-person plural | Q32022732 | Portal:Human settlements |
| Q6581072 | female | Q203516 | birth rate |
| Q618779 | award | Q10815002 | Portal:Family |
| Q28640 | profession | Q8436 | family |
| Q12047083 | professionalism | Q13780930 | worldwide |
| Q19652 | public domain | Q14565199 | right |
| Q101352 | family name | Q542952 | left and right |
| Q113159385 | right-handed person | Q13196750 | left |
| Q2421902 | handedness | Q10764194 | minus sign |
| Q789447 | left-handedness | Q16695773 | WikiProject |
| Q3039938 | right-handedness | Q24025284 | sometimes changes |
| Q73555012 | works protected by copyrights | Q26256810 | topic |
| Q1860 | English | Q2366457 | department |
| Q8229 | Latin script | Q17172850 | voice |
| Q4220917 | film award | Q3348297 | observer |
| Q71887839 | copyrights on works have expired | Q3739104 | natural causes |
| Q82955 | politician | Q11879590 | female given name |
| Q84048852 | female human | Q467 | woman |
| Q3031 | girl | Q188830 | wife |
| Q1196129 | spouse | Q28747937 | history of a city |
| Q3331189 | version, edition or translation | Q4663903 | Wikimedia portal |

Table A.1: Entity expansion blacklist $\mathcal{B}$ used during controlled knowledge graph extraction.

# B   Rule-Based Triple Filtering

We apply a set of rule-based filters composed of seven deterministic rules that identify and remove easily detectable low-quality triples during controlled subgraph extraction. Each rule targets a specific type of undesired pattern, such as identifiers, non-Latin characters, or trivial self-references. The goal is to exclude triples that add little or no meaningful semantic content while keeping computational overhead minimal. Applying these filters early reduces the cost of subsequent processing and enhances the overall quality of the generated dataset. Table B.1 summarizes the implemented rules, their conditions, and corresponding rationales. Each triple $[s, r, o]$, where $s$ is the subject entity, $r$ is the predicate (relation) between two entities, and $o$ is the object entity, is evaluated against the rule set in sequence and discarded as soon as it violates any rule. This rule set was empirically derived by inspecting frequent noise patterns in the Wikidata knowledge base.

# C   Illustrative Examples of the Data Generation Pipeline

This section presents illustrative examples to provide the reader with a clear understanding of the key stages in our data generation pipeline.

| Rule | Condition | Rationale |
|------|-----------|-----------|
| $r_1$ | Predicate does not belong to the blacklist: "Wolfram Language entity code", "Wolfram Language unit code", "Wikidata property", "on focus list of Wikimedia project", "Commons category", "has part(s) of the class", "properties for this type", or "described by source". | Removes predicates that do not convey meaningful semantic relations. |
| $r_2$ | Predicate does not contain the word "ID". | Removes triples with identifier-related predicates, as these contain ID information rather than meaningful semantic relations and thus provide little value for generating coherent text in subsequent stages of the data generation pipeline. |
| $r_3$ | Object does not contain "http://" or "https://". | Removes triples where the object is a URL, as these represent external references rather than conveying semantic content. |
| $r_4$ | Triple does not contain non-Latin script characters (Chinese, Arabic, Persian, Cyrillic, Bopomofo, Katakana, Greek, Bengali, or Hebrew). | Removes triples with non-Latin characters, as mixed scripts can negatively impact the coherence of the corresponding text generated in subsequent stages of data generation. |
| $r_5$ | Subject or object does not start with "Category:", "Template:", "Wikipedia:", or "Portal:". | Removes triples representing organizational metadata that do not convey factual information useful for text generation. |
| $r_6$ | Subject or object does not start with "Q" followed by at least 5 digits. | Removes Wikidata entity identifiers (e.g., Q12308941) that lack human-understandable meaning and cannot be naturally represented in text. |
| $r_7$ | Subject and object are not identical. | Removes self-referential triples that convey trivial information. |

Table B.1: General rule-based filtering criteria applied to knowledge graph triples.

- ["left", "subclass of", "side"]
- ["left", "part of", "left and right"]
- ["left", "Commons category", "Left"]
- ["left", "opposite of", "right"]
- ["left", "instance of", "body relative direction"]

**Figure C.1**: Examples of triples obtained by expanding the entity *left* (Q13196750), which is included in the entity expansion blacklist.

**Triples Removed by Rule-Based Filtering**

- ["Poland", "Wolfram Language entity code", "Entity["Country", "Poland"]"]
- ["association football player", "properties for this type", "DZFoot.com player ID"]
- ["Wikimedia Foundation", "ITU/ISO/IEC object ID", "1.3.6.1.4.1.33298"]
- ["Regionalverband Ruhr", "official website", "https://www.rvr.ruhr/"]
- ["United Kingdom", "demonym", "英国人"]
- ["pneumonia", "topic's main template", "Template:Pneumonia"]
- ["teacher", "category for eponymous categories", "Q59576065"]
- ["Poland", "hashtag", "Poland"]

**Figure C.2**: Examples of triples filtered through our rule-based mechanism to ensure higher-quality knowledge graph extraction.

**Subject–Predicate Pair Uniqueness Violation**

- **Triples:**
  - ["US", "diplomatic relation", "France"]
  - ["US", "diplomatic relation", "Italy"]
- **Invalid Question:**
  - Which country has diplomatic relations with Steve Jobs' origin country?

**Figure C.3**: Example triples violating the uniqueness of subject–predicate pairs. Maintaining this constraint enhances the dataset's utility for potential downstream applications such as multi-hop QA pair generation, where contradictions would otherwise lead to there being no unique answer.

**Triples Retained by Controlled Extraction**

- ["Ladislaus I of Hungary", "native language", "Hungarian"]
- ["Ladislaus I of Hungary", "child", "Irene of Hungary"]
- ["Ladislaus I of Hungary", "member of political party", "politician before the emergence of political parties"]
- ["Ladislaus I of Hungary", "place of burial", "Cathedral Basilica of St. Mary"]
- ["Cathedral Basilica of St. Mary", "country", "Romania"]
- ["Cathedral Basilica of St. Mary", "dedicated to", "Mary"]
- ["Mary", "father", "Joachim"]
- ["Mary", "languages spoken, written or signed", "Imperial Aramaic"]
- ["Mary", "ethnic group", "Jewish people"]

```
▶ ["Mary", "spouse", "Joseph"]
▶ ["Joseph", "given name", "Hovsep"]
▶ ["Joseph", "languages spoken, written or signed", "Aramaic"]
▶ ["Joseph", "field of work", "carpentry"]
▶ ["Joachim", "place of birth", "Tzippori"]
```

**Figure C.4**: Examples of triples preserved by our controlled extraction approach.

**Generated Text**

Ladislaus I of Hungary was a prominent historical figure whose native language was Hungarian. He was the father of Irene of Hungary and is often categorized as a politician before the emergence of political parties. After his death, Ladislaus I was buried in the Cathedral Basilica of St. Mary, located in Romania. This cathedral is dedicated to Mary, a central figure in Christian tradition. Mary, who was of Jewish ethnicity, was the daughter of Joachim, born in Tzippori, and was married to Joseph. She was fluent in Imperial Aramaic, as was Joseph, whose given name was Hovsep and who worked as a carpenter.

**Figure C.5**: Text generated based on the set of triples shown in Figure C.4.

# D   Supplementary Analysis of Dataset Statistics

In this section, we present visualizations of the triple count and token count distributions for the train and test partitions of the datasets summarized in Table 1. The corresponding plots are shown in Figures D.1, D.2, D.3, D.4, and D.5 for KELM, WebNLG+2020, GenWiki-HIQ, CE12k, and CE12k-Relaxed, respectively. Moreover, similar visualizations for the CE-MultiCat-1200, CrossEval-1200, and CrossEval-1500 test sets are provided in Figures D.6, D.7, and D.8, respectively.

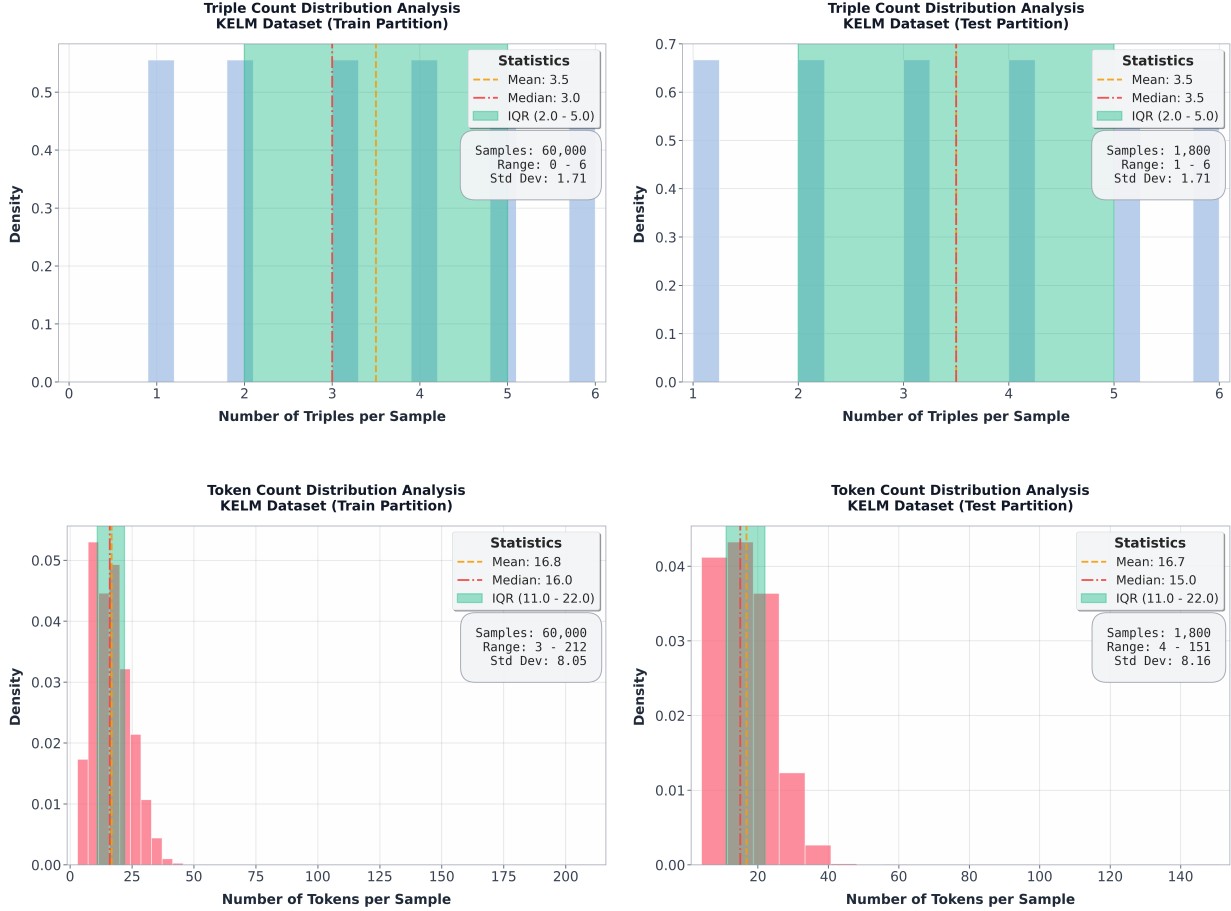

Figure D.1: Triple and token count distribution analyses for the KELM dataset across train and test partitions.

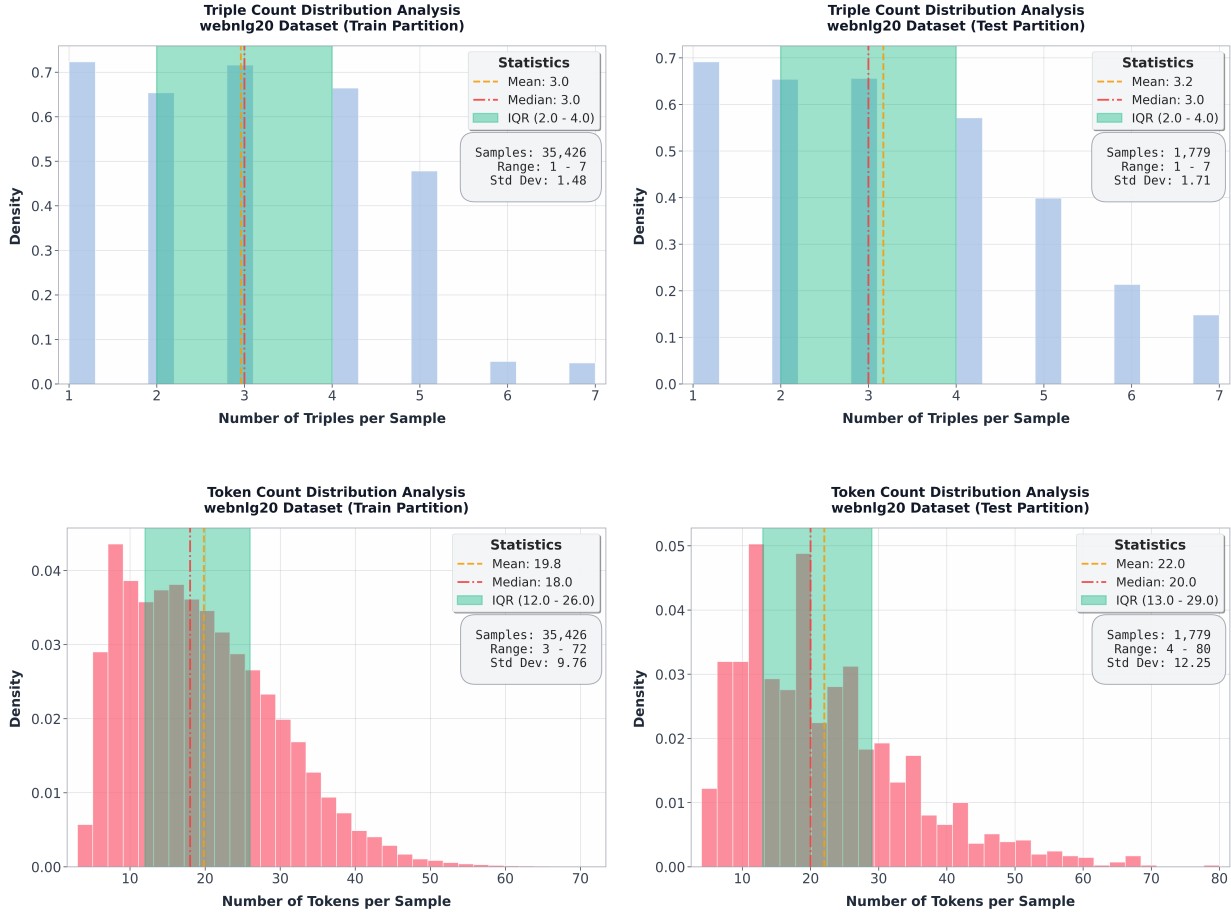

Figure D.2: Triple and token count distribution analyses for the WebNLG+2020 dataset across train and test partitions.

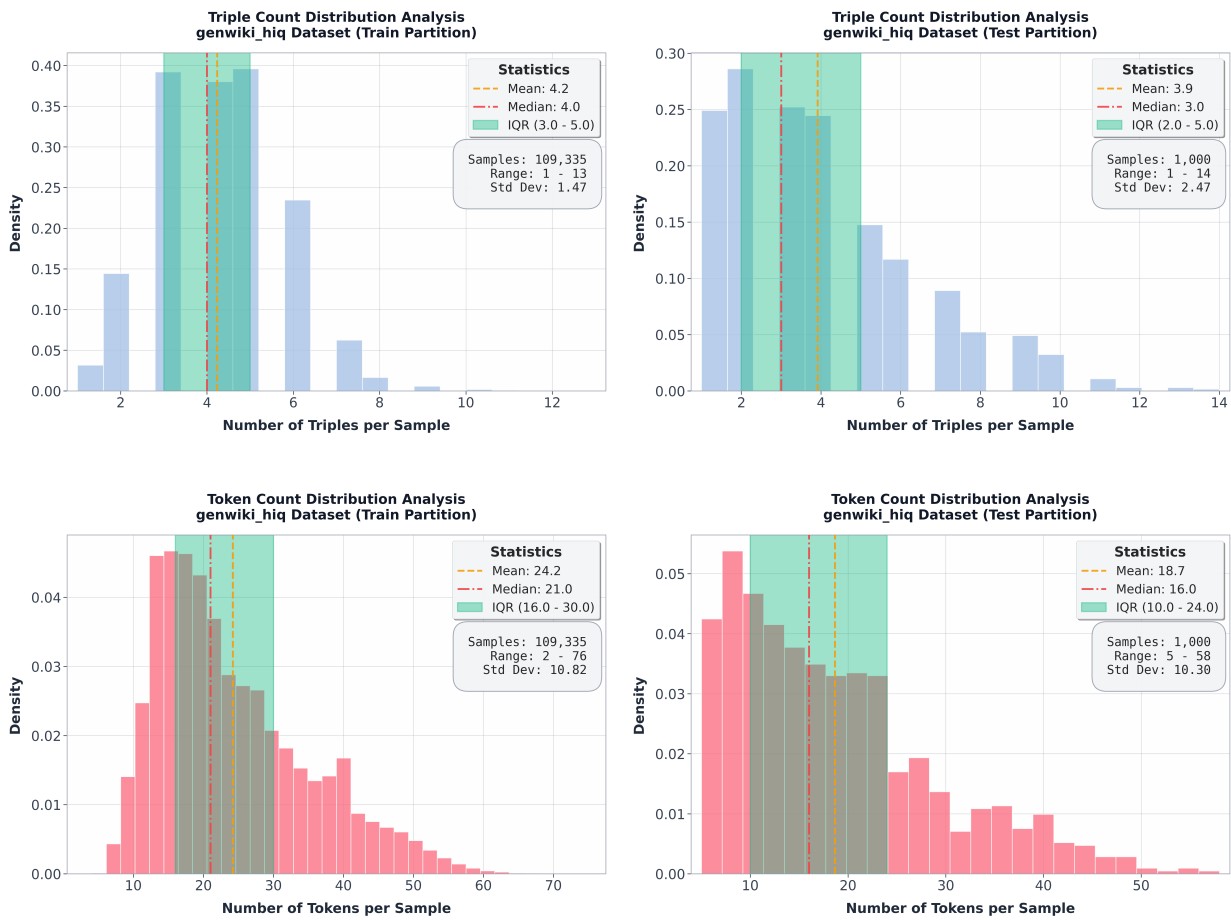

Figure D.3: Triple and token count distribution analyses for the GenWiki-HIQ dataset across train and test partitions.

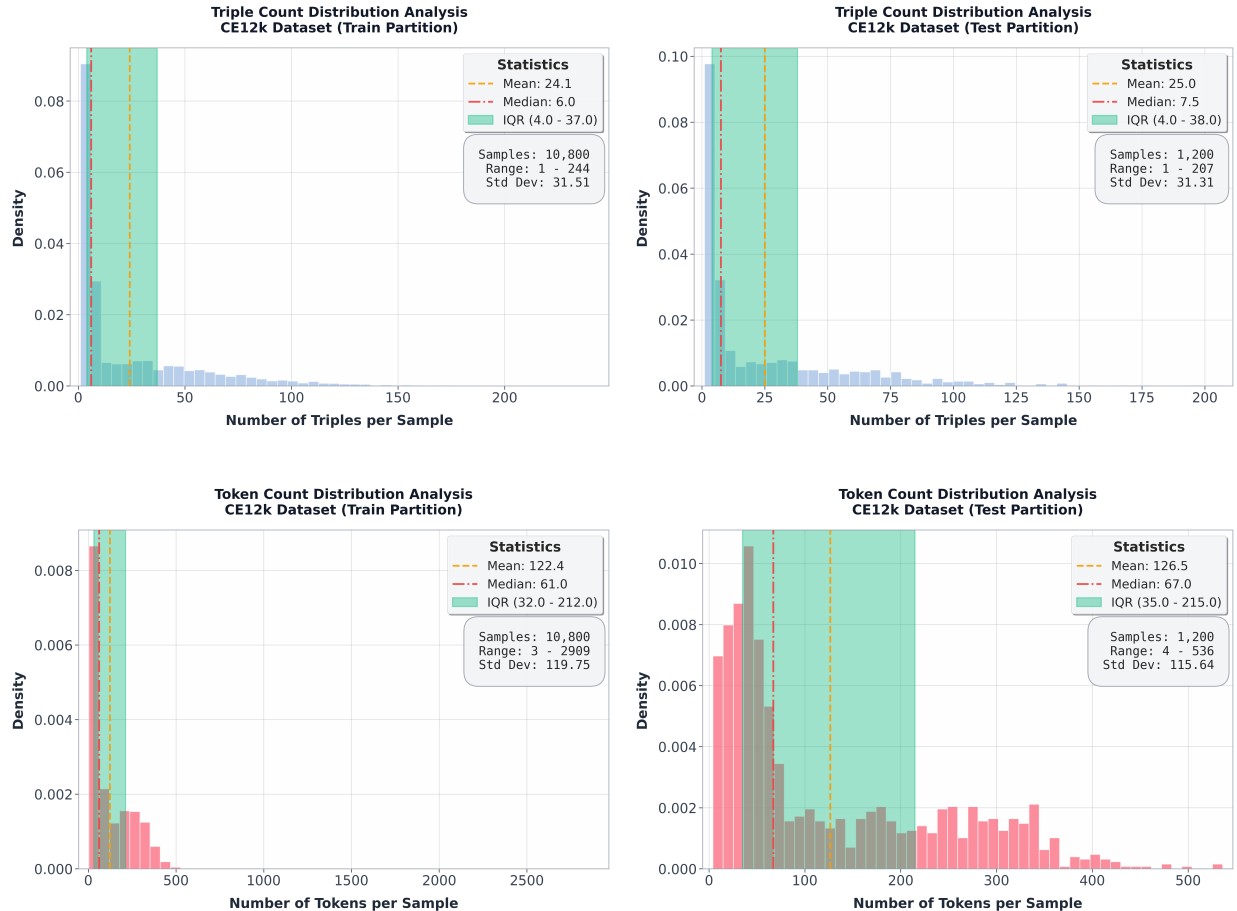

Figure D.4: Triple and token count distribution analyses for the CE12k dataset across train and test partitions.

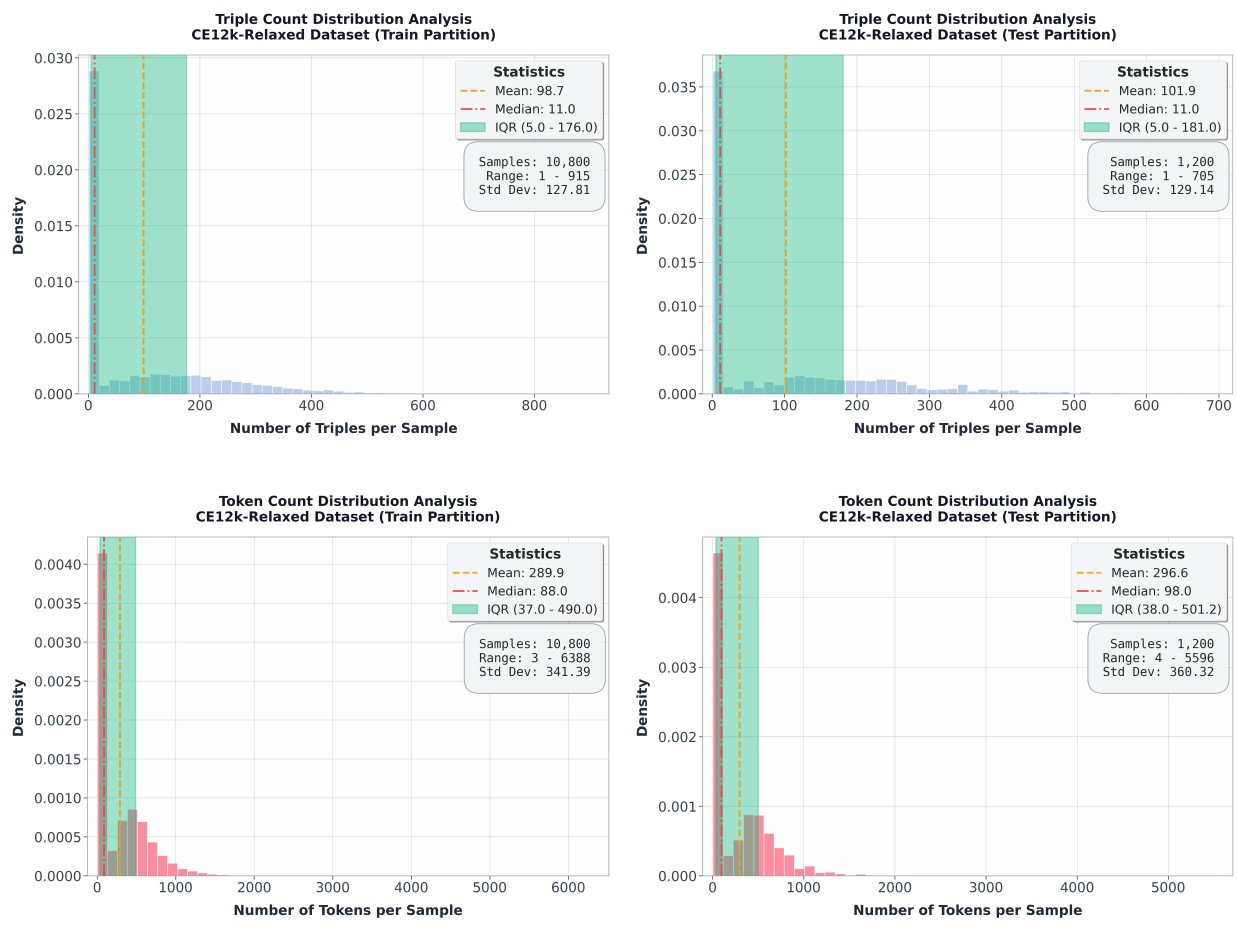

Figure D.5: Triple and token count distribution analyses for the CE12k-Relaxed dataset across train and test partitions.

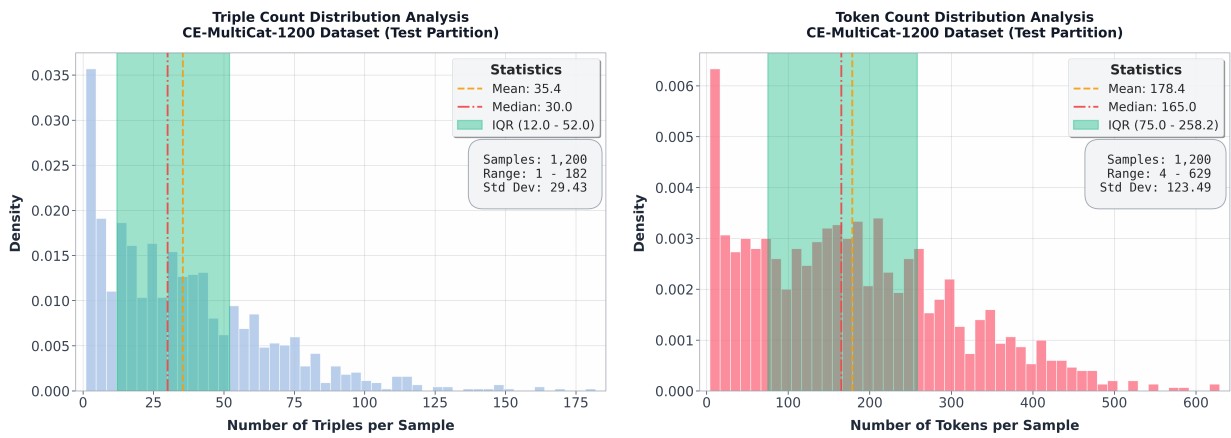

Figure D.6: Triple and token count distributions of the CE-MultiCat-1200 test set.

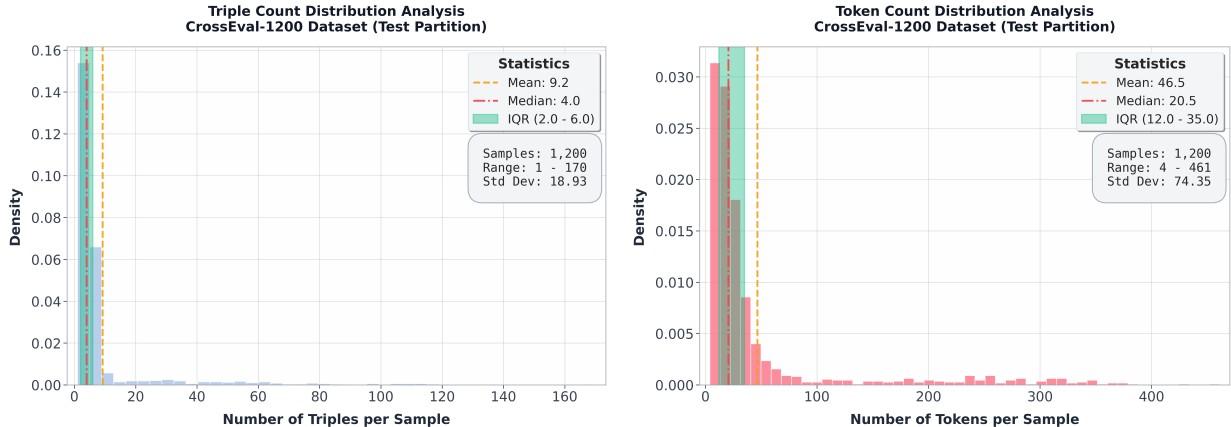

Figure D.7: Triple and token count distributions of the CrossEval-1200 test set.

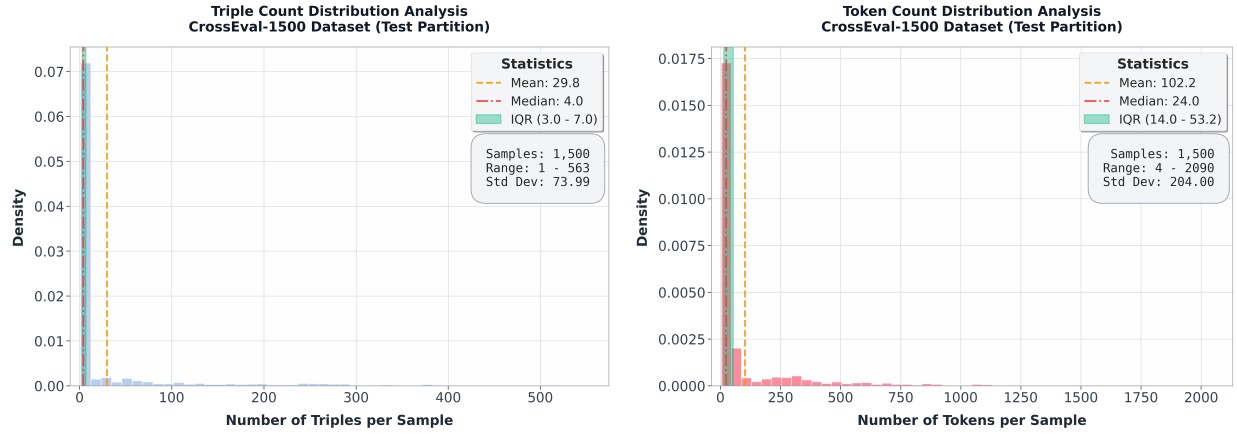

Figure D.8: Triple and token count distributions of the CrossEval-1500 test set.

# E   Task-Specific Prompts

In this section, we present the prompts employed in our work. Specifically, Figure E.1 shows the prompt used for the ChatGPT baseline, while Figure E.2 illustrates the prompt designed for generating descriptive text from the extracted subgraphs. Furthermore, Figure E.3 presents the prompt used for identifying entities whose outgoing triples convey trivial or no information, thereby determining whether they should be added to the entity expansion blacklist.

---

**ChatGPT Baseline Prompt**

```
You are a knowledge graph generator.  Given a text, extract entities and their relationships,
and represent them as a list of triples in the format:  ["subject", "relation", "object"].

Examples:
Example 1:
Text:  Coburg Peak (, ) is the rocky peak rising to 783 m in Erul Heights on Trinity Peninsula
in Graham Land, Antarctica.  It is surmounting Cugnot Ice Piedmont to the northeast.  The peak
is named after the Bulgarian royal house of Coburg (Saxe-Coburg-Gotha), 1887-1946.
Triples:  [["TRINITY PENINSULA", "part of", "GRAHAM LAND"], ["TRINITY PENINSULA", "continent",
"ANTARCTICA"], ["GRAHAM LAND", "continent", "ANTARCTICA"]]
Example 2:
Text:  Harald Kaas (19 May 1868 - 5 December 1953) was a Norwegian architect.  Kaas was born
in Christiania (now Oslo), Norway.  He studied at the Norwegian National Academy of Craft and
Art Industry, then at Baugewerkschule in Eckernförde and finally at Polytechnicum in Munich.
He worked for a couple of years in the Colony of Natal in South Africa.  He was employed by
the Norwegian State Railways from 1908 to 1914, and designed stations on the Arendal Line,
Bergen Line and Solør Line for the company.  Kaas died on 5 December 1953 and was buried on 20
May 1954 at Vår Frelsers gravlund in Oslo.
Triples:  [["HARALD KAAS", "date of birth", "19 MAY 1868"], ["HARALD KAAS", "date of death",
"5 DECEMBER 1953"], ["HARALD KAAS", "employer", "NORWEGIAN STATE RAILWAYS"], ["POLYTECHNICUM",
"headquarters location", "MUNICH"], ["BERGEN LINE", "owned by", "NORWEGIAN STATE RAILWAYS"]]
Example 3:
Text:  Utus Peak (, ) is the rocky peak rising to 1217 m in Trakiya Heights on Trinity
Peninsula in Graham Land, Antarctica.  The peak is named after the ancient Roman town of Utus
in Northern Bulgaria.
Triples:  [["TRAKIYA HEIGHTS", "continent", "ANTARCTICA"], ["TRINITY PENINSULA", "part of",
"GRAHAM LAND"], ["TRINITY PENINSULA", "continent", "ANTARCTICA"], ["GRAHAM LAND", "continent",
"ANTARCTICA"]]

Now, generate the list of triples for the following text:
Text:  {input text}
Triples:
```

**Figure E.1**: Prompt used for the ChatGPT baseline.

**Prompt for Generating Descriptive Text for Each Extracted Subgraph**

```
You are a text generator that reconstructs the original text from a given knowledge
graph.  The knowledge graph is represented as a list of triples in the format:  ["subject",
"relation", "object"].

Your task is to generate a coherent, concise, and natural text that could have been the origin
of the given knowledge graph.  The text should accurately describe the relationships and
entities in the triples, ensuring it is informative and logically structured.

Guidelines:
    1. The generated text can consist of one or more paragraphs, depending on the complexity
       of the triples.

    2. Ensure the text flows naturally, as if it were written by a human.

    3. Include all entities and relationships from the triples.

    4. Avoid adding any information not present in the triples.
Triples:  {input triples}
Text:
```

**Figure E.2**: Prompt used for KG-to-Text generation.

**Prompt for Identifying Entities with Non-Informative Expansions**

```
You are an expert in knowledge graph analysis.  Decide if a batch of triples about an entity
contains only NON-INFORMATIVE knowledge.
Definition
        • NON-INFORMATIVE: trivial, obvious, generic, or vague facts that do not add meaningful
          knowledge.  This includes:
            – Common sense or obvious traits (e.g., humans are mortal, fire is hot)
            – Basic opposites or simple taxonomic facts (e.g., male opposite of female, male
              different from man/masculinity)
            – Overly broad or vague relations that apply to almost any entity (e.g., human has
              effect artificial object, human interacts with environment)
        • INFORMATIVE: specific, distinctive, or non-obvious facts that provide concrete
          knowledge about the entity (e.g., birthplace, achievements, historical events,
          numerical values).
Examples
NON-INFORMATIVE:
["male", "opposite of", "female"]
["human", "has characteristic", "mortality"]
["minus sign", "opposite of", "plus sign"]
["human", "has effect", "artificial object"]
["human", "physically interacts with", "natural environment"]
INFORMATIVE:
["Albert Einstein", "born in", "Ulm"]
["Albert Einstein", "developed", "theory of relativity"]
["Paris", "capital of", "France"]
["Paris", "population", "2,161,000"]
Task
Entity:  {entity_name} ({entity_id})
Triples:
{triples_text}
Question
Are all of these triples NON-INFORMATIVE?
Output (STRICT)
YES (all non-informative)
NO (at least one informative)
```

**Figure E.3**: Prompt for curating the entity expansion blacklist.

## F   Comparison of Generated Outputs on a CE12k Test Set Example

Figure F.1 illustrates a sample from the CE12k test set, showing the ground-truth knowledge graph alongside outputs generated by different methods. We also compute evaluation metrics for this example, with the results presented in Table F.1, offering insight into the results reported in Table 2.

Table F.1: Quantitative comparison of different methods on an example from the CE12k test set, illustrated in Figure F.1. Best results in **bold**, second-best underlined.

| Method | G-BLEU F1 | G-ROUGE F1 | G-BERTScore F1 |
|---|---|---|---|
| OpenIE6 | 23.13 | 20.82 | 76.33 |
| DeepEx | 4.05 | 12.00 | 69.34 |
| PIVE | 57.88 | 65.71 | 83.28 |
| AutoRE | 56.68 | 62.22 | 65.92 |
| ChatGPT | 69.03 | 76.67 | 97.57 |
| GraphRAG | 22.21 | 19.23 | 73.83 |
| LightRAG | 9.75 | 10.77 | 66.81 |
| InvertiTune (Ours) | **1.00** | **1.00** | **1.00** |

**Input Text** $\mathcal{T}$

José Eduardo dos Santos, a citizen of Angola, was a devout Catholic.  He passed away in Barcelona.

**Ground Truth Knowledge Graph** $\mathcal{G}$

[["JOSÉ EDUARDO DOS SANTOS", "religion or worldview", "CATHOLICISM"], ["JOSÉ EDUARDO DOS SANTOS", "place of death", "BARCELONA"], ["JOSÉ EDUARDO DOS SANTOS", "country of citizenship", "ANGOLA"]]

**Knowledge Graph** $\mathcal{G}$ **Generated by OpenIE6**

[["José Eduardo dos Santos", "was", "a devout Catholic"], ["José Eduardo dos Santos", "is a of", "Angola a devout Catholic"], ["Santos", "is a citizen of", "Angola"], ["He", "passed away", "in Barcelona"]]

**Knowledge Graph** $\mathcal{G}$ **Generated by DeepEx**

[["JOSÉ EDUARDO", "dos Santos , a citizen of Angola", "A DEVOUT CATHOLIC"], ["HE", "passed away in", "BARCELONA"]]

**Knowledge Graph** $\mathcal{G}$ **Generated by PIVE**

[["José Eduardo dos Santos", "given name", "José"], ["José Eduardo dos Santos", "citizen of", "Angola"], ["José Eduardo dos Santos", "religion", "Catholic"], ["José Eduardo dos Santos", "place of death", "Barcelona"]]

**Knowledge Graph** $\mathcal{G}$ **Generated by AutoRE**

[["Barcelona", "country", "Angola"], ["Barcelona", "located in the administrative territorial entity", "Angola"], ["José Eduardo dos Santos", "place of death", "Barcelona"], ["José Eduardo dos Santos", "place of birth", "Angola"], ["José Eduardo dos Santos", "country of citizenship", "Angola"], ["José Eduardo dos Santos", "religion", "Catholic"]]

**Knowledge Graph** $\mathcal{G}$ **Generated by ChatGPT**

[["JOSÉ EDUARDO DOS SANTOS", "citizen of", "ANGOLA"], ["JOSÉ EDUARDO DOS SANTOS", "religion", "CATHOLICISM"], ["JOSÉ EDUARDO DOS SANTOS", "place of death", "BARCELONA"]]

**Knowledge Graph** $\mathcal{G}$ **Generated by GraphRAG**

[["JOSÉ EDUARDO DOS SANTOS", "José Eduardo dos Santos was a citizen of Angola", "ANGOLA"], ["JOSÉ EDUARDO DOS SANTOS", "José Eduardo dos Santos passed away in Barcelona", "BARCELONA"]]

**Knowledge Graph** $\mathcal{G}$ **Generated by LightRAG**

[["BARCELONA", "Barcelona is a major city in Spain, establishing a direct geographical link between the two.", "SPAIN"], ["BARCELONA", "José Eduardo dos Santos passed away in Barcelona, establishing a geographical connection between the individual and the city.", "JOSÉ EDUARDO DOS SANTOS"], ["ANGOLA", "José Eduardo dos Santos was a citizen of Angola, indicating a strong national identity associated with the country.", "JOSÉ EDUARDO DOS SANTOS"], ["ANGOLA", "Angola is located in Southern Africa, indicating its geographical classification.", "SOUTHERN AFRICA"], ["JOSÉ EDUARDO DOS SANTOS", "José Eduardo dos Santos is identified as a devout Catholic, highlighting his religious affiliation.", "DEVOUT CATHOLIC"]]

**Knowledge Graph** $\mathcal{G}$ **Generated by InvertiTune**

[["JOSÉ EDUARDO DOS SANTOS", "religion or worldview", "CATHOLICISM"], ["JOSÉ EDUARDO DOS SANTOS", "place of death", "BARCELONA"], ["JOSÉ EDUARDO DOS SANTOS", "country of citizenship", "ANGOLA"]]

Figure F.1: Example outputs from different methods on a sample from the CE12k test set, along with the ground truth.

