# OpenReview forum: "InvertiTune: High-Quality Data Synthesis for Cost-Effective Single-Shot Text-to-Knowledge Graph Generation"
_TMLR — Accepted by TMLR_

### Review · Reviewer_U1Jy · 2026-02-08

**Summary Of Contributions:**

This paper addresses the Test-to-Knowledge-Graph(Text2KG) task by proposing InvertiTune, a framework that inverts the usual pipeline. It first extracts high-quality subgraphs from a large knowledge base (Wikidata) using a controlled, rule-based traversal with noise filtering, and then uses a LLM to generate corresponding text. The resulting (text, KG) pairs are used to perform SFT on lightweight LLM. The authors demonstrates that fine-tuning Qwen2.1-1.5B on CE12k dataset (the 12000 (text, KG) samples generated in this method) significantly outperforms larger non-fine-tuned LLMs and several Text2KG baselines. They also performed an analysis of dataset scaling, suggesting that data quality matters more than data size.

Strengths:
- The described data generation approaches is well motivated by leveraging LLMs strengths (text generation) rather than weakness (strutted extraction). The guideline is clean, well-explained and easy to reproduce.
- The results are strong - it significantly outperforms other approaches on the CE12k test set. The improvements still hold (though smaller) on CrossEval-1200.
- The authors performed thorough experiments to evaluate InvertiTune, covering multiple baselines, metrics, and statistical significance tests.
- It also demonstrated parameter efficiency and single-shot inference, which is practically valuable.
- The authors release the datasets and cross-dataset evaluation, enhance the paper's reliability and reproducibility.

Weaknesses
- The generated text is derived directly from ground-truth KGs, which may introduce distributional bias or task simplification compared to real-world text containing redundancy or ambiguity. It also assumes the LLM generates text perfectly represent the graphs, if it hallucinates or omits some triplets, it will harm the data quality.
- The subgraphs are from the Human category of Wikidata, the generalisation ability to other categories is untested.
- The analysis on the filtering impact is limited. The paper provides no statistics on the amount of triplets getting filtered, and if there is any distribution of relations/entities changes after filtering. There is also no comparison to just use the natural KG.

**Audience:**

Yes

**Audience Explanation:**

This work offers insights on Text2KG generation task with its alternative approaches and cost-efficiency, and also release datasets that could serve as useful benchmark and references for future work.

**Broader Impact Concerns:**

A minor broader impact concern is the potential misuse of the proposed approach for fully automated knowledge base construction without human verification. In addition, since the dataset is generated from a subset of Wikidata, it may reflect existing biases or coverage limitations if used broadly for training. A short disclaimer or reminder to users would be sufficient to address these concerns.

**Claims And Evidence:**

Yes

**Claims Explanation:**

The experiment section is thorough and generally convincing. The authors evaluate against a wide range of baselines, including non-LLM methods, prompt-based LLMs, and SFT-based method. However, it should be noted that the strongest performance gains are observed on CE12k, which is constructed using the same pipeline proposed in the paper. As a result, these results partially reflect in-distribution evaluation. The inclusion of CrossEval-1200, which shows consistent (though smaller) improvements, helps mitigate this concern.

**Requested Changes:**

- All experiments sample come from the Human category in Wikidata. The authors should either provide experiments on at least one additional category, or demonstrate the generalisation abilities on test dataset generated on other categories, or provide justification. (Critical)
- Add some analysis or justification of the subgraph filtering, especially the subject-predicate uniqueness constraint - though it is motivated by downstream QA use cases, it could change the natural distribution of KGs. More discussions and justifications, or experiments comparing the impact of this constraint (e.g. datasets generated with and without this constraint, and the corresponding SFT model performances on the crossEval-1200 dataset) is desired. (Critical)

---

> ### Author Response · Authors · 2026-03-27
>
> We thank the reviewer for the valuable comments and suggestions. We address each point below.
>
> ---
>
> **(1) Text Generation from Ground-Truth KGs and Potential Bias and Coverage Issues**
>
> We thank the reviewer for raising this point.
>
> First, we would like to clarify that although we design several noise filtering rules during subgraph extraction, the resulting KGs may not be perfectly clean. In practice, there may still exist redundancy, noise, or ambiguity in the extracted triples. As a result, the generated texts can also reflect such imperfections, which makes them closer to real-world text rather than overly simplified versions.
>
> Second, our design choice of generating text from KGs is motivated by formulating the task in a way that is potentially more tractable for current LLMs, compared to the reverse direction of extracting KGs from text, which can be more challenging. As noted earlier, the motivation for adopting this automated pipeline is that manually constructing such datasets is not practical. As a result, the generated data may not be completely error-free. In particular, the LLM may occasionally hallucinate content that is not grounded in the KG or omit certain triples, resulting in a text that does not fully represent the underlying structured knowledge.
>
> We partially acknowledge this limitation in the paper by noting that our current pipeline relies on a single model (DeepSeek-V3) for text generation, and that exploring alternative LLMs may further improve the quality and correctness of the generated data.
>
> Future work may explore incorporating automatic consistency checking between the generated text and the KG as a post-processing step.

---

> ### Author Response · Authors · 2026-03-27
>
> **(2) Generalization Beyond the Human Category**
>
> We appreciate the reviewer’s comment on this point.
>
> To address this concern, we extend our evaluation by introducing a new test set, **CE-MultiCat-1200**, consisting of 1,200 samples. In this dataset, for each data sample, the initial entity is sampled from one of four diverse Wikidata categories, namely *Mountain*, *University*, *Film*, and *City*, while deliberately excluding the *Human* category used in CE12k. Each category contributes 300 samples, ensuring a clear distribution shift between the training and test sets. More detailed statistics and construction details of this dataset are provided in the revised version of the paper.
>
> We re-perform two sets of experiments on this new test set. First, we compare InvertiTune against all baselines under this shifted distribution:
>
> | Method                 | G-BL      | G-RO      | G-BS      |
> | ---------------------- | --------- | --------- | --------- |
> | OpenIE6                | 10.74     | 18.86     | 73.42     |
> | DeepEx                 | 5.70      | 10.73     | 44.39     |
> | PIVE                   | 33.52     | 41.22     | 69.21     |
> | AutoRE                 | 26.16     | 30.05     | 69.40     |
> | ChatGPT                | 24.95     | 32.75     | 64.89     |
> | GraphRAG               | 6.01      | 7.67      | 44.19     |
> | LightRAG               | 4.51      | 6.29      | 46.04     |
> | **InvertiTune (Ours)** | **62.20** | **63.38** | **84.00** |
>
> Second, we evaluate cross-dataset generalization by fine-tuning the same backbone model on different training datasets and testing on CE-MultiCat-1200:
>
> | Training Dataset | G-BL      | G-RO      | G-BS      |
> | ---------------- | --------- | --------- | --------- |
> | KELM             | 24.50     | 26.89     | 40.17     |
> | WebNLG+2020      | 16.19     | 20.39     | 53.46     |
> | GenWiki-HIQ      | 14.01     | 17.53     | 35.10     |
> | **CE12k (Ours)** | **62.20** | **63.38** | **84.00** |
>
> Across both evaluations, InvertiTune (i.e., the Qwen2.5-1.5B Instruct model fine-tuned on the training set of CE12k) consistently outperforms all baselines by a substantial margin. These results demonstrate that the improvements are not limited to the Human category and generalize to other domains with different distributions.

---

> ### Author Response · Authors · 2026-04-02
>
> **(3) Analysis and Justification of the Filtering Strategy**
>
> We thank the reviewer for highlighting the importance of analyzing the impact of this design choice in the subgraph filtering stage.
>
> To address this point, we include an ablation study in the revised manuscript that directly evaluates its effect. Specifically, we construct a variant of our dataset, denoted **CE12k-Relaxed**, by disabling the subject–predicate pair uniqueness constraint while keeping all other components of the pipeline unchanged. We then fine-tune the same model on both CE12k and CE12k-Relaxed under identical training settings.
>
> We evaluate the resulting models on multiple test sets, including KELM, GenWiki-HIQ, WebNLG+2020, and **CrossEval-1500**, where the latter extends CrossEval-1200 by incorporating additional samples from the CE12k-Relaxed test split to enable a more controlled and balanced comparison. The results are summarized below:
>
> | Test Dataset   | Training      | G-BL  | G-RO  | G-BS  |
> | -------------- | ------------- | ----- | ----- | ----- |
> | KELM           | CE12k-Relaxed | 55.00 | 62.55 | 81.17 |
> |                | CE12k         | 53.87 | 62.21 | 81.71 |
> | WebNLG+2020    | CE12k-Relaxed | 32.88 | 41.24 | 79.66 |
> |                | CE12k         | 38.55 | 47.59 | 86.16 |
> | GenWiki-HIQ    | CE12k-Relaxed | 29.30 | 33.73 | 78.74 |
> |                | CE12k         | 29.83 | 34.87 | 77.49 |
> | CrossEval-1500 | CE12k-Relaxed | 49.11 | 53.30 | 80.16 |
> |                | CE12k         | 52.66 | 57.38 | 83.57 |
>
> As shown above, the model trained on CE12k performs comparably to, and in some cases better than, the model trained on CE12k-Relaxed. In particular, performance is similar on KELM and GenWiki-HIQ, while CE12k achieves modest improvements on WebNLG+2020 and CrossEval-1500.
>
> These findings indicate that applying this constraint does not negatively impact the effectiveness of the resulting training data for Text2KG, while supporting downstream applications such as question answering.
>
> We have added this analysis, along with detailed experimental results, to the revised version of the paper.
>
> **(4) Broader Impact and Potential Misuse**
>
> We thank the reviewer for raising this important point.
>
> To address this concern, we have added a concise disclaimer to the Limitations section of the revised manuscript. Specifically, we clarify that, as with any automated data generation approach, the resulting data may contain errors or omissions in the absence of human verification. We also note that, since the dataset is constructed from a subset of Wikidata, it may reflect biases or coverage limitations of that source.
>
> In addition, we explicitly highlight that these aspects should be taken into account when using the dataset, particularly in settings involving fully automated knowledge base construction without human oversight.

---

### Review · Reviewer_AbMn · 2026-02-24

**Summary Of Contributions:**

### Summary
The authors propose a benchmark and data-generation framework for the Text-to-Knowledge-Graph (Text2KG) task. The framework introduces a newly created dataset (CE12k) and a supervised fine-tuning setup. The authors justify the dataset primarily through within-dataset and cross-dataset model comparisons, where their fine-tuned model outperforms several baselines.

### Strengths

**(S1 - novelty).**
The idea of inverting the generation process (KG → text) for dataset construction is convinving and has the potential to be impactful if the resulting datasets generalize beyond curated settings.

**(S2 — clarity and relevance wrt research objectives tackled).**
The introduction and the related work section are very well written. The authors clearly explain the state-of-the-art and motivate why their work is relevant. Without being an expert in the KG field, the argumentation appears meaningful and convincing in terms of potential relevance for the community.

**(S3 — significance through statistical tests).**
The authors include statistical tests (paired Wilcoxon signed-rank test) in their experiments, making performance gaps statistically significant.

**(S4 — quality).**
The writing style is generally of high quality and easy to follow. I appreciate the comprehensive description of the data creation pipeline. While I am not yet convinced that all design choices are optimal, the authors describe them clearly and transparently.

### Weaknesses

**(W1 — design choices in the dataset creation pipeline).**
The authors design certain operators to yield “high-quality KGs", including omitting triplets of low additional information. While I understand that these choices may produce more meaningful samples, I am not certain whether applying these filters leads to a more artificial dataset distribution compared to real-world knowledge graphs. Could the authors please share their thoughts here?

**(W2 — design of experiments).**
I appreciate the attempt to measure across-dataset generalization, which is important to address concerns raised in (W1). However, instead of fusing datasets, it would have been stronger to fine-tune a model solely on CE12k and evaluate directly on external datasets. With the fused version, it remains unclear whether performance gains stem from specific subgroups, e.g., CE12k samples themselves.

**(W3 — clarity wrt intention of experiments).**
If I understand correctly, the authors demonstrate performance gains of their tuned model to indirectly show that their created dataset is meaningful. Making this intention more explicit earlier in the experimental section would help, since the narrative shifts quickly from benchmark design to model comparison.

**Audience:**

Yes

**Audience Explanation:**

Knowledge graphs are inherently valuable because structured representations enable a wide range of downstream applications, e.g., providing meaningful context for LLM-based systems or grounding reasoning models on verifiable steps. Consequently, methods that improve the generation of such structured data are timely and relevant to the community.

**Broader Impact Concerns:**

no broader impact concerns.

**Claims And Evidence:**

No

**Claims Explanation:**

“No” may be somewhat strong here; however, a binary choice was required. In practice, my concern relates to two specific claims made in the introduction for which supporting citations or demonstrations should be added.

In the introduction the authors write:

> However, the reliance on extensive prompting makes these approaches computationally expensive, and early mistakes may propagate through later stages, degrading the graph quality.

and

> Given the complexity of the task for current models, the resulting datasets may contain errors or incomplete information.

Taken together, these statements create the impression that existing methods are inherently error-prone, which forms a central part of the motivation for InvertiTune. However, the manuscript currently does not provide direct empirical demonstrations or citations that substantiate these weaknesses.

Adding supporting references, empirical observations, or concrete examples would strengthen the motivation and help justify the necessity of the proposed approach.

**Requested Changes:**

### Critical
* (W1): It would be important to clarify that the data creation pipeline does not introduce artificial graph structures that deviate substantially from real Wikidata subgraphs. Providing additional rationale or evidence in the rebuttal would help address this concern.
* (W2): As discussed above, a stronger cross-domain evaluation would be a key ingredient for demonstrating real-world relevance. It would strengthen the paper to fine-tune on CE12k and evaluate directly on external datasets.

### Optional
* (W3): The experimental section could benefit from clearer motivation and signposting to help readers understand the intent behind each experiment.
* The choice of fixed dataset-generation hyperparameters (m=3, k=2) raises questions about structural diversity. Could this lead to overly homogeneous graphs in terms of hops and neighborhood structure? It may be worth discussing whether sampling the number of neighbors and hop depth dynamically during extraction would improve diversity and realism.

---

> ### Author Response · Authors · 2026-04-02
>
> We thank the reviewer for the thoughtful comments and suggestions. We address each point below.
>
> **(1) Impact of Filtering and Potential Distribution Shift**
>
> We thank the reviewer for raising this point. We address this concern from multiple perspectives.
>
> First, we would like to emphasize that our work introduces a framework that tightly couples a controlled data generation pipeline with supervised fine-tuning (SFT), rather than proposing a standalone dataset. In this context, “dataset quality” refers to how suitable the generated data is for training Text2KG models. Accordingly, we evaluate the generated data through its effectiveness in the downstream Text2KG task, where we show that the model fine-tuned on our dataset consistently outperforms both larger non-fine-tuned LLMs and models trained on existing datasets. Based on the reviewer’s helpful comment, we also realized that the term “high-quality” may introduce ambiguity; therefore, we have removed this term from the title and revised its usage throughout the paper to better reflect the intended meaning.
>
> Second, the main goal of this work is to make available models more capable of performing the Text2KG task, that is, generating the corresponding knowledge graph when given a text. In this sense, our objective is not to construct a dataset whose extracted subgraphs exactly match the distribution of real-world knowledge graphs. In practice, real-world knowledge graphs often contain some level of noise or triples with limited informational value; therefore, considering this, a certain degree of distribution shift due to filtering can be plausible. However, for the purpose of supervised training, the more important property is the quality of the correspondence between the text and the target KG. Our pipeline is designed to strengthen this correspondence by inverting the task: instead of asking an LLM to generate the KG from text, which can be more challenging, we start from a KG and ask the LLM to generate the corresponding text.
>
> Third, we analyze the effect of the filtering strategy through an ablation study included in the revised manuscript. Specifically, we construct a variant of our dataset by disabling one of the filtering mechanisms while keeping all other components of the pipeline unchanged, and fine-tune the same model under identical settings. The results show that the model trained on the original dataset performs comparably to, and in some cases slightly better than, the model trained on the relaxed version. The detailed results across multiple test sets are reported below. This indicates that applying this filtering step does not negatively impact the effectiveness of the resulting training data, while in some cases leading to modest improvements. We have added this analysis and the corresponding results to the revised version of the paper.
>
> | Test Dataset   | Training      | G-BL  | G-RO  | G-BS  |
> | -------------- | ------------- | ----- | ----- | ----- |
> | KELM           | CE12k-Relaxed | 55.00 | 62.55 | 81.17 |
> |                | CE12k         | 53.87 | 62.21 | 81.71 |
> | WebNLG+2020    | CE12k-Relaxed | 32.88 | 41.24 | 79.66 |
> |                | CE12k         | 38.55 | 47.59 | 86.16 |
> | GenWiki-HIQ    | CE12k-Relaxed | 29.30 | 33.73 | 78.74 |
> |                | CE12k         | 29.83 | 34.87 | 77.49 |
> | CrossEval-1500 | CE12k-Relaxed | 49.11 | 53.30 | 80.16 |
> |                | CE12k         | 52.66 | 57.38 | 83.57 |

---

> ### Author Response · Authors · 2026-04-02
>
> **(2) Cross-Dataset Evaluation and Generalization**
>
> We appreciate the reviewer’s insightful comment regarding the evaluation design and the concern about dataset fusion.
>
> To address this point, we extend our experiments by performing evaluation in a setting that more closely aligns with the reviewer’s suggestion, namely training on CE12k and evaluating on a test set with a different distribution.
>
> In particular, we introduce a new test set, **CE-MultiCat-1200**, where the initial entity is sampled from four distinct Wikidata categories (*Mountain*, *University*, *Film*, and *City*), while excluding the *Human* category used in CE12k. This construction ensures a clear distribution shift between training and evaluation. We then evaluate the model fine-tuned on CE12k on this test set, without any dataset fusion.
>
> The results are summarized below:
>
> | Method                 | G-BL      | G-RO      | G-BS      |
> | ---------------------- | --------- | --------- | --------- |
> | OpenIE6                | 10.74     | 18.86     | 73.42     |
> | DeepEx                 | 5.70      | 10.73     | 44.39     |
> | PIVE                   | 33.52     | 41.22     | 69.21     |
> | AutoRE                 | 26.16     | 30.05     | 69.40     |
> | ChatGPT                | 24.95     | 32.75     | 64.89     |
> | GraphRAG               | 6.01      | 7.67      | 44.19     |
> | LightRAG               | 4.51      | 6.29      | 46.04     |
> | **InvertiTune (Ours)** | **62.20** | **63.38** | **84.00** |
>
> Additionally, we evaluate cross-dataset generalization by fine-tuning the same backbone model on different training datasets and testing on CE-MultiCat-1200:
>
> | Training Dataset | G-BL      | G-RO      | G-BS      |
> | ---------------- | --------- | --------- | --------- |
> | KELM             | 24.50     | 26.89     | 40.17     |
> | WebNLG+2020      | 16.19     | 20.39     | 53.46     |
> | GenWiki-HIQ      | 14.01     | 17.53     | 35.10     |
> | **CE12k (Ours)** | **62.20** | **63.38** | **84.00** |
>
> These results follow the evaluation setup described above and show that the performance gains are not driven by dataset fusion or specific subgroups. Instead, the model trained on CE12k consistently generalizes to other categories and outperforms alternatives by a substantial margin.
>
> We have added these experiments and clarifications to the revised version of the paper to strengthen the evaluation design and improve clarity.
>
> **(3) Clarification of Experimental Intent**
>
> We thank the reviewer for this helpful observation. To improve clarity, we revised the beginning of the *Experiments* section to explicitly state that the goal of our experiments is to assess the effectiveness of the proposed data generation pipeline by evaluating whether the generated datasets can support supervised fine-tuning (SFT) for the Text2KG task, thereby serving as an indirect measure of their usefulness.
>
> This clarification has been added to the revised version of the paper.
>
> **(4) Supporting Motivation Claims**
>
> We thank the reviewer for this helpful comment. We have added supporting references in the revised version of the paper to better justify the statements in the introduction.

---

> > ### Comment · Reviewer_AbMn · 2026-04-07
> > **Response to authors**
> >
> > Dear authors,
> >
> > thank you for your rebuttal.
> >
> > **(2) Cross-Dataset Evaluation and Generalization**
> > This is great. The experiment clearly demonstrates the benefits of fine-tuning models on the CE12k dataset.
> >
> > **(1) Potential Distribution Shift**
> > I am still not fully convinced that potential distribution shift is negligible. I agree that the crux lies in generating suitable data for the Text2KG task. Naturally, one would assume that suitable data should stem from the data distribution encountered at test time (application time). However, I also agree that suitability can be somewhat assessed through the presented results, and your strong rebuttal on point (2) clearly supports this.
> >
> > Thus, my concerns are resolved, and I will adjust my recommendation accordingly.

---

### Review · Reviewer_osG8 · 2026-02-25

**Summary Of Contributions:**

This paper proposes InvertiTune, a framework for single-shot Text-to-Knowledge Graph (Text2KG) generation based on a controlled data synthesis pipeline and SFT. Instead of extracting knowledge graphs from text, the method inverts the process by first extracting semantically coherent subgraphs from a knowledge base and then generating corresponding textual descriptions using an LLM. The generated dataset (CE12k) is used to fine-tune a lightweight model (Qwen2.5-1.5B), enabling efficient one-pass KG generation. Experiments show that the proposed method significantly outperforms prompting-based Text2KG approaches and larger unfine-tuned models on CE12k and CrossEval-1200 benchmarks. The results suggest that high-quality synthetic data can enable strong performance with relatively small models.

**Audience:**

Yes

**Audience Explanation:**

I think some individuals who worked on Text2KG and data synthesis may be interested in this paper.

**Claims And Evidence:**

Yes

**Claims Explanation:**

I am not very professional in this field. From my perspective, the paper provides strong empirical results, but some claims are insufficiently justified.

1. The paper claims the generated dataset is high-quality and realistic. But I think there is some evidence missing: no comparison with current datasets and no human evaluation (not sure if needed in this field).
2. Performance is great, but only mainly evaluated on CE12K.
3. Generalization experiments are limited; can try more models and text generators.

**Requested Changes:**

1. Provide a direct evaluation of the dataset quality
2. Efficiency measurements should be added, such as latency comparison..
3. Strengthen generalization evaluation

Refer to `Are the claims made in the submission supported by accurate, convincing and clear evidence?` for more details.

---

> ### Author Response · Authors · 2026-03-26
>
> We thank the reviewer for the valuable comments and suggestions. We address each point below.
>
> **(1) Dataset Quality Evaluation**
>
> We thank the reviewer for raising this point. Below, we first clarify what we mean by “dataset quality” and then describe the changes we made to avoid any unintended ambiguity.
>
> First, we would like to emphasize that our work introduces a framework that tightly couples a controlled data generation pipeline with supervised fine-tuning (SFT), rather than proposing a standalone dataset. As such, the quality of the generated data is evaluated through its effectiveness in the downstream Text2KG task. In the paper, we conduct extensive experiments demonstrating that the model fine-tuned on our generated dataset consistently outperforms both larger non-fine-tuned LLMs and models trained on existing datasets. We aim to provide a practical indication of the usefulness of the generated data for the intended task through this evaluation.
>
> Second, we relate dataset quality to how well it supports learning under more representative and challenging Text2KG settings. As discussed in the paper, the existing datasets contain knowledge graphs with an average of fewer than 5 triples and corresponding texts with an average of fewer than 25 tokens. Such settings result in relatively trivial instances where KG extraction can be performed reliably by general-purpose LLMs without task-specific fine-tuning, or even by humans. As a result, these datasets can be considered somewhat toy-like and do not adequately reflect real-world scenarios, where inputs are typically longer and, consequently, the corresponding knowledge graphs to be extracted are larger. In contrast, our generated dataset contains texts exceeding 120 tokens on average and the corresponding knowledge graphs containing more than 24 triples on average, which is more aligned with practical scenarios and leads to a more challenging Text2KG setting. Furthermore, our generation pipeline allows controllable scaling of graph size and text length through parameters (e.g., number of hops and neighbors), enabling the construction of datasets with even longer texts and the corresponding larger knowledge graphs.
>
> By “dataset quality” in this work, we refer to how suitable the generated data is for training Text2KG models under these settings. Following this clarification, and based on the reviewer’s helpful comment, we realized that the term “high-quality” may introduce ambiguity. To improve clarity, we have removed this term from the title and revised its usage throughout the paper to better reflect the intended meaning.
>
> **(2) Performance Mainly Evaluated on CE12K**
>
> Thank you for bringing this to our attention. To address this valuable comment, we extend our evaluation beyond the CE12k test set by introducing an additional test set with a different data distribution.
>
> Specifically, we construct CE-MultiCat-1200, where the initial entity is sampled from four distinct Wikidata categories, namely Mountain, University, Film, and City, while excluding the Human category used in CE12k. This design ensures a clear distribution shift between training and evaluation. The dataset consists of 1,200 samples, with 300 instances per category, and subgraphs are extracted using $m = 4$ neighboring nodes and $k = 6$ hops.
>
> This setting enables evaluation under a different and more challenging scenario. As shown below, InvertiTune consistently outperforms all baselines across all evaluation metrics by a substantial margin:
>
> | Method                 | G-BL      | G-RO      | G-BS      |
> | ---------------------- | --------- | --------- | --------- |
> | OpenIE6                | 10.74     | 18.86     | 73.42     |
> | DeepEx                 | 5.70      | 10.73     | 44.39     |
> | PIVE                   | 33.52     | 41.22     | 69.21     |
> | AutoRE                 | 26.16     | 30.05     | 69.40     |
> | ChatGPT                | 24.95     | 32.75     | 64.89     |
> | GraphRAG               | 6.01      | 7.67      | 44.19     |
> | LightRAG               | 4.51      | 6.29      | 46.04     |
> | **InvertiTune (Ours)** | **62.20** | **63.38** | **84.00** |
>
> These results indicate that the observed performance is not limited to CE12k and generalizes beyond the training distribution. We have added these results, along with a detailed description of the CE-MultiCat-1200 test set, to the revised version of the paper.

---

> ### Author Response · Authors · 2026-03-26
>
> **(3) Efficiency Measurements (Latency Comparison)**
>
> Thank you for this valuable comment. To address this point, we report the average inference time per sample for each method as a reference point for latency.
>
> The measured latency results are summarized below:
>
> | Method                 | Avg. Inference Time (s) |
> | ---------------------- | ----------------------- |
> | OpenIE6                | 52.71                   |
> | DeepEx                 | 126.33                  |
> | PIVE                   | 4.65                |
> | AutoRE                 | 19.56                   |
> | ChatGPT                | 7.18                    |
> | GraphRAG               | 42.11                   |
> | LightRAG               | 24.10                   |
> | InvertiTune (Ours) | 9.05                    |
>
> These results show that our approach achieves competitive latency while maintaining strong performance.
>
> We have added these latency measurements to the revised version of the paper.
>
> **(4) Strengthening Generalization Evaluation**
>
> We appreciate the reviewer’s suggestion on strengthening the generalization evaluation. In response, we have enriched the *Cross-Dataset Generalization* section of the paper.
>
> More precisely, to provide a stricter assessment of generalization, we extend the evaluation to CE-MultiCat-1200, a test set whose distribution differs from all training sets used in this comparison. Unlike CrossEval-1200, where part of the test samples share a similar distribution with the training data of each model, CE-MultiCat-1200 presents a more challenging out-of-distribution scenario for all models.
>
> As shown below, the model fine-tuned on CE12k continues to outperform all counterparts by a substantial margin:
>
> | Training Dataset | G-BL      | G-RO      | G-BS      |
> | ---------------- | --------- | --------- | --------- |
> | KELM             | 24.50     | 26.89     | 40.17     |
> | WebNLG+2020      | 16.19     | 20.39     | 53.46     |
> | GenWiki-HIQ      | 14.01     | 17.53     | 35.10     |
> | **CE12k (Ours)** | **62.20** | **63.38** | **84.00** |
>
> The performance gap is even more pronounced compared to the CrossEval-1200 results, supporting supervised fine-tuning on CE12k, which is generated via our introduced data generation pipeline, as a more effective way to obtain models with stronger transferability.
>
> We have added these results, along with a detailed discussion, to the revised version of the paper.

---

> > ### Comment · Reviewer_osG8 · 2026-03-26
> >
> > Thanks for your reply! I do not have further question.

---

### Decision · Action_Editor_x5Qo · 2026-04-16

**Recommendation:** Accept with minor revision

**Additional Comments:**

One reviewer still has some concerns about the model's generalisation ability: 1) There is no additional evidence or justification is provided to support generalisation beyond the Human Category of Wikidata. 2) There is no further analysis about the improvements on CrossEval-1200. It would be important to address these concerns in the revision.

**Audience:**

Yes

**Audience Explanation:**

The paper presents a well-motivated and practically valuable approach to Text-to-Knowledge-Graph (Text2KG) generation, which is an important task in related areas.

**Claims And Evidence:**

Yes

**Claims Explanation:**

The experimental evaluation is thorough, and the improvements (particularly in terms of parameter efficiency and single-shot inference) are compelling

---

> ### Author Response · Authors · 2026-05-08
>
> Thank you for raising these important points. We would like to clarify them as follows.
>
> 1) Generalization beyond the Human category
>
> To address the concern regarding generalization beyond the Human category in Wikidata, we created an additional out-of-distribution test set, CE-MultiCat-1200, specifically designed to evaluate generalization across diverse entity domains. In this benchmark, the initial entity e0 is sampled from four distinct Wikidata categories: Mountain, University, Film, and City, deliberately excluding the Human category to avoid overlap with the original distribution.
>
> We repeated the full experimental comparison between InvertiTune and all baselines on this benchmark, and the results are now included in Table 3 of the revised manuscript. The results show that InvertiTune consistently outperforms all competing baselines across all three evaluation metrics by a substantial margin, demonstrating that its superior performance generalizes effectively to unseen entity domains.
>
> 2) Performance gains on CrossEval-1200
>
> Regarding the observed improvements on CrossEval-1200, one key contributing factor is the structural differences between the training datasets. As shown in Table 1, the CE12k dataset used for training InvertiTune contains samples with substantially more realistic complexity compared to alternative training datasets such as KELM, WebNLG+2020, and GenWiki-HIQ, where the average number of triples per sample is typically fewer than five.
>
> Fine-tuning the same backbone model on simpler datasets encourages learning short, low-complexity extraction patterns, which limits the model’s ability to generate larger knowledge graphs when presented with longer, more realistic input texts. In contrast, CE12k was intentionally constructed to include more complex graph structures, enabling the model to better learn longer-range relational dependencies.
>
> To further support the results observed on CrossEval-1200 and assess whether this generalization behavior consistently holds under another out-of-distribution setting, we repeated the comparison on CE-MultiCat-1200. The superiority of InvertiTune remains consistent on this benchmark as well, further reinforcing that the observed gains reflect stronger generalization rather than dataset-specific effects.